# GoT: Unleashing Reasoning Capability of MLLM for Visual Generation and Editing

**Rongyao Fang**[1][*] **Chengqi Duan**[3][*] **Kun Wang**[4] **Linjiang Huang**[1] **Hao Li**[1,5] **Hao Tian**[4]
**Shilin Yan** **Weihao Yu**[1] **Xingyu Zeng**[4,6] **Jifeng Dai**[5] **Xihui Liu**[3][†] **Hongsheng Li**[1,2][†]

[1]CUHK MMLab  [2]CPII under InnoHK  [3]HKU  [4]SenseTime  [5]Shanghai AI Lab  [6]SUAT

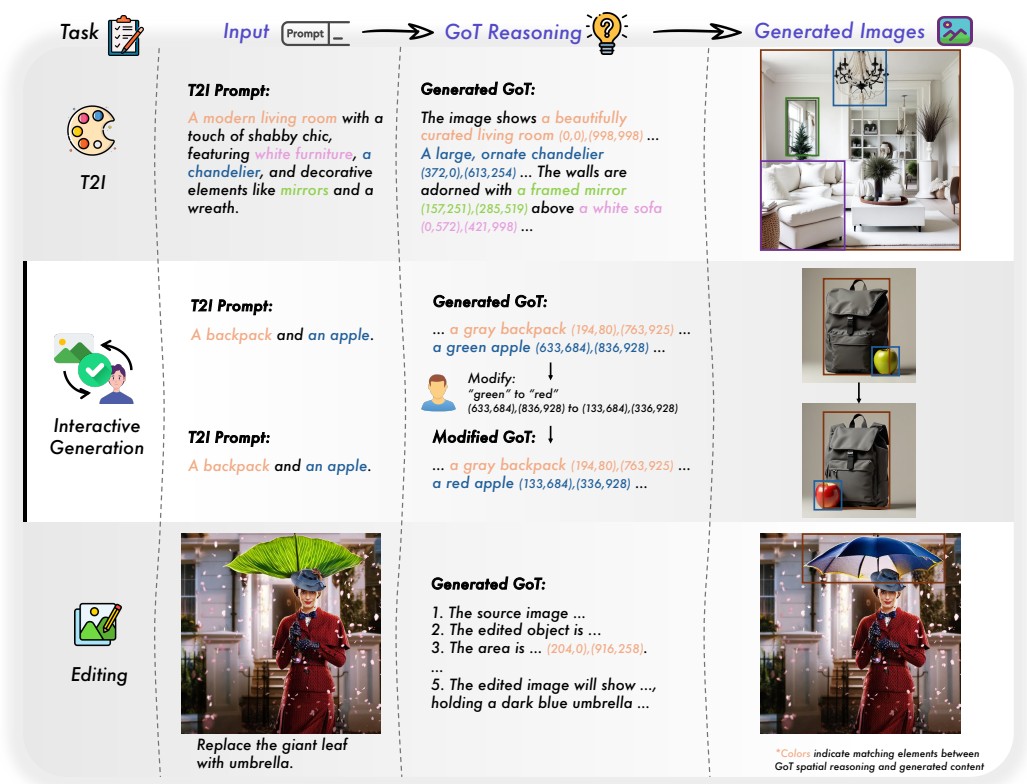

Figure 1: **Generation Chain-of-Thought (GoT) with Semantic-Spatial Reasoning.** Our approach transforms input prompts into explicit reasoning chains with coordinates (middle), which guides vivid image generation and precise editing (right). This reasoning-based generation paradigm unifies spatial understanding across visual tasks: semantically-grounded visual generation (top), controllable interactive generation (middle), and localized image editing (bottom).

## Abstract

Current image generation and editing methods primarily process textual prompts as direct inputs without explicit reasoning about visual composition or operational steps. We present Generation Chain-of-Thought (GoT), a novel paradigm that empowers a Multimodal Large Language Model (MLLM) to first generate an explicit, structured reasoning chain in natural language—detailing semantic relationships,

---

[*]Equal Contribution
[†]Corresponding Authors

39th Conference on Neural Information Processing Systems (NeurIPS 2025).

object attributes, and, crucially, precise spatial coordinates—before any image synthesis occurs. This intermediate reasoning output directly guides the subsequent visual generation or editing process. This approach transforms conventional text-to-image generation and editing into a reasoning-guided framework that analyzes semantic relationships and spatial arrangements. We define the formulation of GoT and construct large-scale GoT datasets containing over **9M** samples with detailed reasoning chains capturing semantic-spatial relationships. To leverage the advantages of GoT, we implement a unified framework that integrates Qwen2.5-VL for reasoning chain generation with an end-to-end diffusion model enhanced by our novel Semantic-Spatial Guidance Module. Experiments show our GoT framework achieves excellent performance on both generation and editing tasks, with significant improvements over baselines. Additionally, our approach enables interactive visual generation, allowing users to explicitly modify reasoning steps for precise image adjustments. GoT pioneers a new direction for reasoning-driven visual generation and editing, producing images that better align with human intent. We will release our datasets and models to facilitate future research.

# 1   Introduction

Language provides the primary interface for expressing human intent in visual content generation. Traditional image generation systems [37, 6, 21], particularly diffusion models, process textual prompts by mapping semantic concepts to visual elements without explicit reasoning. These approaches struggle with complex scenes requiring precise spatial arrangements and object interactions that humans naturally consider when constructing scenes. Meanwhile, multimodal large language models (MLLMs) [3, 2, 25] excel at sophisticated reasoning tasks, including analyzing semantic structures, inferring relationships, grounding visual concepts, and processing detailed contexts through explicit reasoning chains. This gap between MLLMs' advanced reasoning capabilities and the limited reasoning in current generation systems raises a key question: How can we integrate the reasoning mechanisms that have revolutionized language understanding into visual generation and editing?

Prior work attempted to leverage LLMs for image generation from different perspectives. One line of research [23, 55] leverages LLMs as text encoders for better prompt interpretation. However, the reasoning capabilities of LLMs are not introduced. Another line of work develops multimodal LLMs to unify understanding and generation [44, 47, 50, 7]. Although they present unified models for different tasks, there is no evidence that generation benefits from strong understanding and reasoning abilities of the models. They merely combine independent tasks rather than truly fusing language reasoning with visual generation. Additionally, layout-based methods like GLIGEN [22], LayoutGPT [9], and RPG [52] incorporate LLMs for layout planning and diffusion models for layout-guided generation. However, these methods treat planning and generation as separate stages rather than integrating reasoning throughout the end-to-end process. Consequently, current image generation methods lack reasoning capabilities, emphasizing the need for a framework that seamlessly combines reasoning with visual generation and editing.

Inspired by chain-of-thought (CoT) reasoning of the LLMs, we introduce Generation Chain-of-Thought (GoT), a novel paradigm that enables visual generation to first output step-by-step reasoning in natural language before producing images. However, implementing GoT poses two significant challenges. First, different from CoT in LLMs, the reasoning chain for visual generation and editing requires both semantic and spatial information. It requires a new formulation and collecting training data in this new format. Second, existing diffusion-based models cannot leverage explicit language reasoning chains during visual generation. We need to design a framework supporting end-to-end language reasoning and visual generation.

To address the first challenge, we formulate GoT as a multimodal reasoning chain that integrates semantic and spatial analyses to enhance image generation and editing tasks. For visual generation, GoT provides precise control over object layout, relationships, and attributes, while for editing, it leverages semantic and spatial understanding to decompose user requests into coherent grounding and modification steps. We utilize advanced MLLMs and LLMs to construct complex annotation pipelines, which capture semantic-spatial interactions across diverse visual contexts. We assembled extensive datasets comprising 8.4M images for text-to-image generation (from Laion-Aesthetics [39], JourneyDB [41], and FLUX [21]) and 920K examples for image editing (from OmniEdit [48] and SEED-Edit-Multiturn [12]). This computationally intensive effort produced the first large-scale dataset of reasoning chains for image generation and editing.

To tackle the second challenge of architecture design supporting reasoning and generation, we construct a unified end-to-end framework. Our GoT framework integrates the reasoning capabilities of MLLMs with the high-fidelity generation qualities of diffusion models. The proposed framework leverages an MLLM to generate reasoning steps and visual tokens, providing explicit guidance that incorporates semantic relationships and spatial configurations. This guidance flows into our novel Semantic-Spatial Guidance Module (SSGM), which conditions the diffusion process to ensure that generated images are closely guided by the reasoning process. This design supports end-to-end training and inference for visual generation and editing guided by explicit reasoning chains.

By effectively integrating reasoning into visual generation, our GoT framework demonstrates significant improvements in both text-to-image generation quality and image editing accuracy. Additionally, GoT enables interactive generation, allowing users to control the generated image by directly modifying the explicit reasoning process according to their preferences. These advantages represent a substantial advancement in reasoning-guided visual synthesis.

The main contributions can be summarized as follows:

- We propose Generation Chain-of-Thought (GoT), a paradigm where an MLLM first generates an explicit semantic-spatial reasoning chain—comprising textual descriptions of objects, their attributes, inter-relationships, and precise geometric coordinates—to plan visual content before generation or editing.

- We define the formulation of semantic and spatial reasoning chains for visual generation and editing, and constructed the first large-scale GoT datasets, encompassing 8.4M image generation, 920K image editing samples. Creating this dataset, with its semantic-spatial annotations derived from complex MLLM-driven annotation pipelines, consumed over 3000 NVIDIA A100 GPU days.

- We develop a unified end-to-end framework that leverages multimodal language models and diffusion models, with a novel Semantic-Spatial Guidance Module that ensures generated images follow the reasoning process.

- Our experimental results demonstrate significant improvements in both text-to-image generation and editing.

## 2 Related Work

### 2.1 Diffusion Models

Diffusion models have revolutionized visual content creation. Early approaches [36, 30] demonstrated this paradigm's potential, while Stable Diffusion [37] improved efficiency through latent space compression. Recent models [35, 38, 32, 6, 21] have further advanced photorealism through architectural innovations and larger-scale training. Various efforts to extend diffusion models' capabilities include controllable generation methods [54, 28] and instruction-based editing frameworks [5, 40]. While some researchers have explored unifying vision tasks [11, 8], these primarily focus on traditional computer vision tasks rather than general image generation. Despite these advances, current models typically process prompts through direct mapping, using text encoders like CLIP [33] or T5 [34] to condition the diffusion process via cross-attention [45]. This approach treats text as a static representation without explicit reasoning about scene composition or object relationships. The fundamental limitation becomes evident when generating complex scenes with multiple objects and specific spatial arrangements, necessitating more sophisticated reasoning-based approaches.

### 2.2 Large Language Models and Reasoning

Large Language Models (LLMs) have demonstrated remarkable reasoning capabilities through chain-of-thought (CoT)[49], enabling complex problem decomposition. This paradigm extends to MLLMs [1, 2], which integrate visual and textual understanding. Some advanced works [25, 31] have enhanced spatial understanding by grounding textual concepts to image regions, enabling analysis of object relationships. Despite these capabilities, MLLMs remain underutilized for visual generation. While models like Chameleon [44] and Emu2 [43] incorporate image generation, they lack mechanisms to decompose user intent into explicit semantic-spatial reasoning steps that guide the synthesis process. More recently, [15] explored applying CoT-inspired strategies to autoregressive image generation, focusing on verifying and reinforcing the iterative token prediction steps of the generation model itself, which differs from our GoT paradigm where an explicit, structured semantic-spatial reasoning chain with coordinates is generated as a guide for image synthesis.

## 2.3 Layout-guided Image Generation and Editing

Recent research has explored layout-guided approaches for spatial control in visual synthesis. GLI-GEN [22] incorporated bounding boxes through gated cross-attention layers, enhancing object placement. LayoutGPT [9] proposed a two-stage pipeline converting text into scene layouts before generation. RPG [52] advanced this through recurrent planning, alternating between layout refinement and synthesis. SmartEdit [17] adapts the LLaVA [26] model to specialize in image editing tasks. FlexEdit [29] employs an MLLM to comprehend the image content, mask, and user instructions. Despite these advances, existing approaches treat layouts as static constraints or sequential plans generated before synthesis, disconnecting spatial planning from generation.

## 3 Generation Chain-of-Thought (GoT)

During visual generation and editing, humans naturally reason about object relationships and spatial arrangements. In contrast, most current models process prompts without explicit reasoning, making it difficult to interpret complex human intentions for generating scenes with detailed object relationships and spatial configurations.

Motivated by chain-of-thought (CoT) in language models, we propose Generation Chain-of-Thought (GoT), shifting the visual generation from direct mapping to a reasoning-guided process. Unlike language generation, which operates primarily within a semantic space, visual generation requires an integrated understanding of both semantic relationships and spatial configurations. To address this complexity, GoT employs a multi-modal reasoning formulation that bridges conceptual understanding and spatial reasoning. This formulation incorporates explicit coordinate information in format (x1,y1),(x2,y2) with range [0,1000], ensuring precise management of visual element placement. This unified semantic-spatial reasoning chain enables fine-grained control of object placement, attributes, and inter-object relationships, ultimately supporting robust and coherent visual generation.

To illustrate GoT's formulation, Fig. 1 presents examples of both text-to-image generation and editing tasks. For text-to-image, GoT generates a detailed reasoning chain specifying precise coordinates of elements. This explicit spatial reasoning enables proper arrangement of all constituents while maintaining their semantic relationships, resulting in a coherent and visually appealing composition.

The image editing example in Fig. 1 demonstrates how GoT handles manipulation tasks through structured reasoning. When tasked with *replace the giant leaf with an umbrella*, GoT first analyzes the scene and then plans edits with precise coordinates. Finally, GoT describes the edited image. This decomposition into sequential steps with explicit spatial reasoning streamlines complex manipulations, contrasting with traditional editing methods that lack spatial awareness and reasoning.

GoT endows image generation and editing with reasoning benefits. By decomposing complex instructions into clearly defined, sequential steps, GoT delivers results that more accurately fulfill human requests. Its transparent process explains the intermediate reasoning behind each change and enables both image generation and editing within a unified system.

Implementing GoT requires two key components:

- **A Comprehensive Dataset**: This dataset must consist of detailed reasoning chains that align with visual content, capturing both semantic relationships and spatial configurations. Such data provide the necessary foundation for the reasoning process.

- **A Compatible Visual Generation Model**: The model needs to accommodate chain input to integrate semantic analysis and spatial reasoning, ensuring effective execution of the reasoning steps derived from the dataset.

In the following sections, we elaborate on these components and discuss how they contribute to the robust performance of the GoT framework.

## 4 GoT Dataset: Semantic-Spatial Reasoning Chains for Visual Generation and Editing

Based on the formulation presented previously, we construct large-scale training datasets using advanced LLMs and MLLMs. Our GoT dataset features meticulously crafted semantic-spatial reasoning chains for generation/editing tasks, each sample containing instructions, reasoning chain annotations, and corresponding images. Construction requires careful design of task-specific annotation pipelines to ensure quality. The prompts used in the pipelines are attached in Appendix Sec. E.

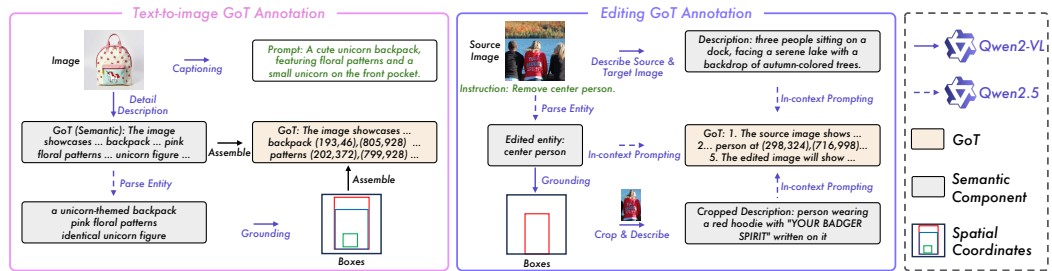

Figure 2: **GoT Dataset Construction Process. Left:** Text-to-image GoT annotation pipeline that labels detailed GoT with semantic content and spatial coordinates. **Right:** Editing GoT annotation pipeline that processes source image, target image, and instruction to generate entity-aware reasoning GoT with precise spatial grounding. Both pipelines leverage Qwen2-VL [46] and Qwen2.5 [51] models for various stages of the annotation process.

## 4.1 Automated Data Creation Pipeline

As illustrated in Fig. 2, our annotation pipeline demonstrates the multiple stages of processing required to generate these high-quality annotations. For text-to-image, we utilize Qwen2-VL [46] to generate concise prompts that serve as text-to-image generation prompts and detailed visual descriptions that form the semantic component of GoT. Qwen2.5 [51] then performs object entity extraction, followed by Qwen2-VL establishing spatial relationships through object grounding. The detailed visual descriptions merged with precise object groundings together constitute the complete GoT annotation for text-to-image generation.

For the image editing pipeline, we employ Qwen2-VL to generate comprehensive descriptions of source and target images, precisely localize editing regions through bounding boxes, and generate detailed descriptions of edited objects after cropping. We then leverage Qwen2.5 with carefully designed in-context prompting to synthesize coherent GoT reasoning chains, ensuring logical flow and completeness of the editing process. From this pipeline, we derive concise editing instructions as editing inputs while using the detailed semantic-spatial reasoning steps as GoT annotations. For the complex multi-turn editing dataset, we developed a related but more sophisticated protocol with Qwen2-VL and Qwen2.5 to obtain intricate step-by-step reasoning chains with multiple spatial coordinates and transformation descriptions, capturing complex editing sequences.

## 4.2 Dataset Construction

For text-to-image generation, we construct dataset from three sources: Laion-Aesthetics-High-Resolution (LAHR) [39] with 3.77M samples filtered for images larger than 512 pixels, JourneyDB [41] with 4.09M samples, and 600K FLUX.1-generated [21] images using LAHR prompts.

For the single-turn image editing dataset, we build on OmniEdit [48], a premier open-source image editing dataset with high-fidelity images, processing 736,691 samples covering editing operations (addition, removal, swap, changing expression/color/weather/lighting, and style transfer). The multi-turn image editing dataset is built upon SEED-Edit-Multiturn [12], resulting in 180,190 samples.

## 5 GoT Framework: Reasoning-guided Visual Generation and Editing

We present the GoT framework, a novel architecture that integrates a semantic-spatial aware Multi-modal Large Language Model (MLLM) with a multi-guided diffusion model. The core structural innovation lies in its unified, end-to-end design where an explicit reasoning chain (GoT) is a structured intermediate representation, directly bridging the MLLM's reasoning capabilities with the diffusion model's synthesis process. This design ensures generated images precisely follow logical reasoning steps, allowing detailed control over both semantic content and spatial relationships.

### 5.1 Semantic-Spatial MLLM Design

Our framework utilizes a state-of-the-art MLLM, Qwen2.5-VL-3B, as the backbone, chosen for its outstanding visual understanding and grounding capabilities. This MLLM functions as a reasoning engine, handling both generation and editing tasks through a unified architecture.

As illustrated in Fig. 3, the process begins with task-specific inputs. For editing tasks, reference images are processed through the vision encoder. For both generation and editing, the MLLM first produces the detailed GoT reasoning chain, which captures object attributes, relationships, intended modifications, and bounding box information.

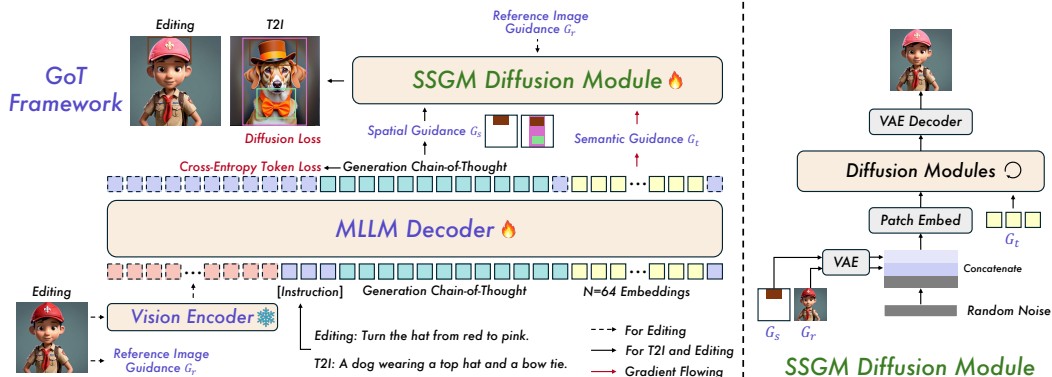

Figure 3: **GoT Framework with Semantic-Spatial Guidance. Left:** Our dual-task framework handling both text-to-image generation (T2I) and image editing. **Right:** The SSGM Diffusion Module, which combines spatial layouts guidance $G_s$, reference image guidance $G_r$, and semantic guidance $G_t$ to generate the final image with precise content and spatial control.

A key aspect of our approach is the generation of semantic guidance $G_t$ for the diffusion model. To achieve this, we introduce a sequence of $N = 64$ dedicated learnable embeddings. Following the generation of the GoT reasoning chain, the MLLM's decoder, which inherently employs causal attention, processes these $N$ learnable embeddings. The sequence of $N$ output vectors from the MLLM decoder, corresponding to these initial learnable embeddings and now richly conditioned by the preceding GoT reasoning chain, forms the semantic guidance $G_t$. These $G_t$ embeddings thus encapsulate detailed semantic information derived directly from the GoT, ready to steer the diffusion process. Concurrently, the spatial guidance $G_s$ is derived by parsing and converting the explicit coordinate information present in the generated GoT reasoning chains.

This semantic-spatial aware design enables the MLLM to direct the SSGM Diffusion Module with precise control over content and layout. During training, the MLLM receives supervision through two pathways: cross-entropy loss on GoT reasoning tokens and gradient signals backpropagated from the end-to-end SSGM diffusion module through semantic guidance $G_t$.

## 5.2 Semantic-spatial Guided Diffusion Generation

The end-to-end diffusion module builds upon SDXL's [32] architecture, incorporating an innovative triple-guidance mechanism. At the heart of this is our Semantic-Spatial Guidance Module (SSGM), a novel architectural component specifically designed to interpret the structured outputs of the GoT (both semantic and spatial) and seamlessly inject these directives into the diffusion module.

In SSGM, the semantic guidance pathway enhances the diffusion model by channeling $N = 64$ MLLM-generated embeddings $G_t$ through cross-attention layers. This represents a significant structural departure from conventional methods: instead of relying on generic text encoder embeddings (e.g., from CLIP), our architecture uses $G_t$—embeddings directly reasoned and generated by the MLLM based on the explicit GoT. This structural choice provides a more nuanced, contextually grounded, and GoT-aligned semantic control signal for the diffusion process.

For spatial guidance in SSGM, we extract coordinate information from the generated GoT to create color-coded masks where each object or editing region receives a distinct color based on a predefined order in the GoT sequence. These colored masks are processed through a VAE encoder [18] and averaged to produce spatial latent features $G_s$, which are concatenated with the diffusion model's latent representations, enabling precise spatial control during both generation and editing tasks.

A crucial structural aspect of the SSGM is its ability to synergistically leverage both the semantic guidance $G_t$ and the spatial guidance $G_s$, which are notably derived from the *same unified GoT reasoning chain*. This co-derivation and joint integration within SSGM ensure a high degree of coherence between the generated content's semantics ('what') and its spatial arrangement ('where'), a hallmark of our framework's design.

Following InstructPix2Pix [5], we incorporate reference image guidance as the third SSGM pathway. For editing tasks, the source image serves as a reference, while for text-to-image generation, we use a black reference image for architectural consistency. This design enables a seamless transition between generation and editing tasks without architectural modifications. All references are processed through the VAE encoder to extract visual features $G_r$.

| Method | Architecture | Overall | Single Obj. | Two Obj. | Counting | Colors | Position | Attr. Binding |
|---|---|---|---|---|---|---|---|---|
| *Frozen Text Encoder Mapping Methods* | | | | | | | | |
| SDv1.5 [37] | Unet+CLIP | 0.43 | 0.97 | 0.38 | 0.35 | 0.76 | 0.04 | 0.06 |
| SDv2.1 [37] | Unet+CLIP | 0.50 | 0.98 | 0.51 | 0.44 | **0.85** | 0.07 | 0.17 |
| SD-XL [32] | Unet+CLIP | 0.55 | 0.98 | **0.74** | 0.39 | **0.85** | 0.15 | 0.23 |
| DALLE-2 [35] | Unet+CLIP | 0.52 | 0.94 | 0.66 | 0.49 | 0.77 | 0.10 | 0.19 |
| SD3 (d=24) [6] | MMDIT+CLIP+T5 | 0.62 | 0.98 | **0.74** | 0.63 | 0.67 | 0.34 | 0.36 |
| *LLMs/MLLMs Enhanced Methods* | | | | | | | | |
| LayoutGPT [9] | Unet+Llama | 0.41 | 0.97 | 0.51 | 0.26 | 0.56 | 0.11 | 0.07 |
| LlamaGen [42] | Autoregressive | 0.32 | 0.71 | 0.34 | 0.21 | 0.58 | 0.07 | 0.04 |
| Chameleon [44] | Autoregressive | 0.39 | - | - | - | - | - | - |
| LWM [24] | Autoregressive | 0.47 | 0.93 | 0.41 | 0.46 | 0.79 | 0.09 | 0.15 |
| SEED-X [13] | Unet+Llama | 0.49 | 0.97 | 0.58 | 0.26 | 0.80 | 0.19 | 0.14 |
| Emu3-Gen [47] | Autoregressive | 0.54 | 0.98 | 0.71 | 0.34 | 0.81 | 0.17 | 0.21 |
| Janus [50] | Autoregressive | 0.61 | 0.97 | 0.68 | 0.30 | 0.84 | 0.46 | **0.42** |
| JanusFlow [27] | Autoregressive | 0.63 | 0.97 | 0.59 | 0.45 | 0.83 | **0.53** | **0.42** |
| **GoT Framework** | Unet+Qwen2.5-VL | **0.64** | **0.99** | 0.69 | **0.67** | **0.85** | 0.34 | 0.27 |

Table 1: Evaluation of text-to-image generation on GenEval benchmark [14]. Obj.: Object. Attr.: Attribution.

## 5.3 Guidance Strategy

We employ a classifier-free guidance strategy integrating semantic, spatial, and reference image guidance. During diffusion, the score estimation $\varepsilon_\theta$ is calculated as a weighted combination of conditional and unconditional scores. Let $\varepsilon_0 = \varepsilon_\theta(z_t, \varnothing, \varnothing, \varnothing)$ denote the unconditional score estimate, $\varepsilon_r = \varepsilon_\theta(z_t, \varnothing, \varnothing, G_r)$ the estimate conditioned only on reference image features $G_r$, $\varepsilon_{t,r} = \varepsilon_\theta(z_t, G_t, \varnothing, G_r)$ the estimate conditioned on semantic guidance $G_t$ and reference features $G_r$, and $\varepsilon_{t,s,r} = \varepsilon_\theta(z_t, G_t, G_s, G_r)$ the estimate conditioned on semantic guidance $G_t$, spatial guidance $G_s$, and reference features $G_r$. The combined score is then:

$$\varepsilon_\theta = \varepsilon_0 + \alpha_t(\varepsilon_{t,r} - \varepsilon_r) + \alpha_s(\varepsilon_{t,s,r} - \varepsilon_{t,r}) + \alpha_r(\varepsilon_r - \varepsilon_0) \tag{1}$$

Here, $z_t$ is the noisy latent. Guidance scales $\alpha_t$, $\alpha_s$, and $\alpha_r$ control the strength of each respective guidance type (semantic, spatial, and reference), while $\varnothing$ denotes null conditioning. During training, we randomly sample conditioning combinations with a probability of 5%, excluding the fully-conditioned case $\varepsilon_{t,s,r}$, to enhance robustness. Guidance parameters are introduced in Sec. 6.

## 6 Experiments

We evaluate GoT framework on text-to-image generation, interactive image generation, and image editing. Experiments show quantitative improvements and qualitative benefits of our reasoning-guided approach, with ablation studies validating our design choices.

### 6.1 Text-to-Image Generation

#### 6.1.1 Quantitative Results

Tab. 1 presents a evaluation of text-to-image generation (T2I) on GenEval [14]. The comparison spans two main categories of models: those employing frozen text encoders for direct prompt-to-image generation (primarily diffusion-based approaches) and those leveraging LLMs or MLLMs to enhance the generation process. On T2I task, GoT framework adopts $\alpha_t = 7.5$ and $\alpha_s = 4.0$, and more discussions on $\alpha$ tuning are shown in Appendix Sec. C.2. The results show our framework achieves the highest overall score of 0.64, outperforming both frozen text encoder methods and LLM/MLLM-enhanced approaches. GoT excels particularly in single object (0.99), counting tasks (0.67), and color tasks (0.85), demonstrating the effectiveness of our reasoning-guided generation paradigm. While methods like JanusFlow [27] perform better in position and attribute binding tasks, GoT's balanced performance across all metrics validates that incorporating explicit reasoning mechanisms enhances compositional generation abilities.

Among the LLM/MLLM-enhanced methods, our approach outperforms recent systems like Janus [50] and JanusFlow [27] in overall performance despite their advantages in specific areas. This suggests that while autoregressive models excel in certain spatial tasks, our GoT framework's structured reasoning provides more consistent performance across diverse generation requirements.

#### 6.1.2 Qualitative Results

In addition to the outstanding compositional text-to-image generation capability, GoT framework also exhibits high generation quality. In Fig. 4, we showcase the generation results of our model across a

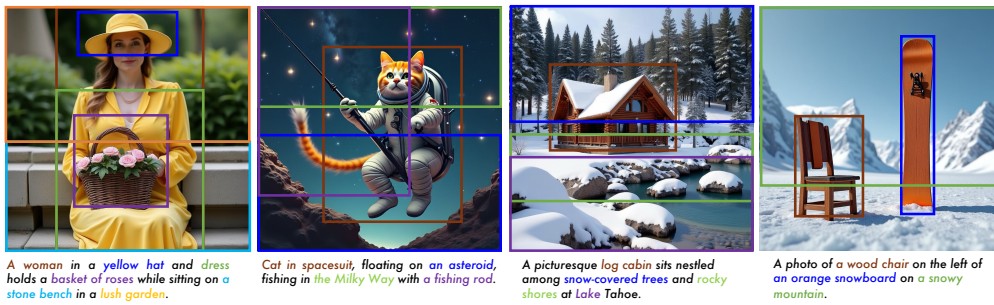

A woman in a *yellow hat* and *dress* holds a *basket of roses* while sitting on a *stone bench* in a *lush garden*.

*Cat in spacesuit*, floating on *an asteroid*, fishing in *the Milky Way* with *a fishing rod*.

A picturesque *log cabin* sits nestled among *snow-covered trees* and *rocky shores* at *Lake* Tahoe.

A photo of a *wood chair* on the left of *an orange snowboard* on *a snowy mountain*.

Figure 4: Text-to-Image samples generated by our model. The GoT framework can plan object placement based on the input caption and generate highly aligned and aesthetic images accordingly.

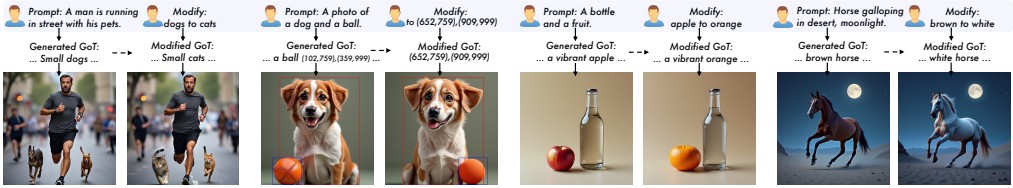

Figure 5: Samples on interactive generation with GoT framework. By modifying GoT content (description and bounding box position), user can customize their text-to-image process with: 1. Object replacement 2. Object position adjustment 3. Object attribute modification.

diverse set of prompts. We present samples from compositional prompts containing multiple objects, incorporating object attributes, relationships, and relative spatial positions. Our model effectively plans the placement of different objects, producing coherent and aesthetically pleasing images.

## 6.2 Interactive Generation

In our experiments, we further demonstrate the interactive capabilities of the GoT framework, illustrated in Fig. 5. This approach enables user control over the generation process by modifying the GoT content, including both textual descriptions and bounding box positions. Users can customize their text-to-image generation through three primary interaction types: object replacement, object position adjustment, and object attribute modification. The examples showcase how the framework maintains overall scene coherence while precisely implementing the requested changes. This flexibility provides an interpretable and manipulable interface for text-to-image generation that traditional black-box systems lack, allowing precise control over the output without requiring expertise.

## 6.3 Image Editing

### 6.3.1 Quantitative Results

As shown in Tab. 2, we evaluate our GoT framework against state-of-the-art image editing methods across multiple benchmarks. On Emu-Edit benchmark [40], GoT framework achieves the highest scores for both CLIP-I (0.864) and CLIP-T (0.276) metrics, outperforming previous methods including CosXL-Edit [4] and

| Method | Params. | Emu-Edit | | ImagenHub | Reason-Edit |
| --- | --- | --- | --- | --- | --- |
| | | CLIP-I | CLIP-T | GPT-4o Eval. | GPT-4o Eval. |
| IP2P [5] | 0.9B+0.1B | 0.834 | 0.219 | 0.308 | 0.286 |
| MagicBrush [53] | 0.9B+0.1B | 0.838 | 0.222 | 0.513 | 0.334 |
| MGIE [10] | 0.9B+7B | 0.783 | 0.253 | 0.392 | 0.264 |
| Emu-Edit [40] | - | 0.859 | 0.231 | - | - |
| SEED-X [13] | 2.8B+14B | 0.825 | 0.272 | 0.166 | 0.239 |
| SmartEdit† [17] | 0.9B+7B | - | - | - | **0.572** |
| CosXL-Edit [4] | - | 0.860 | 0.274 | 0.464 | 0.325 |
| **GoT Framework** | 2.8B+3B | **0.864** | **0.276** | **0.533** | 0.561 |

Table 2: Quantitative comparison on image editing benchmarks. † denotes that SmartEdit mainly supports removing and replacing operation and is not designed for general editing operations.

Emu-Edit [40]. Since CLIP-I and CLIP-T cannot fully reflect editing accuracy, we also evaluated using GPT-4o [1], which aligns better with human evaluation [19]. On ImagenHub [20], our approach attains the highest score of 0.533. On the reasoning-based Reason-Edit benchmark [17], our model achieves a strong score of 0.561, second only to SmartEdit (0.572) [17], which is specially designed for reasoning removing and replacing operations. This demonstrates our method's strong editing ability, especially in complex reasoning settings. GoT framework shows consistently superior performance while maintaining competitive parameter efficiency (2.8B+3B) compared to approaches like SEED-X (2.8B+14B) [13]. In the editing task, GoT framework adopts $\alpha_t = 4.0$, $\alpha_s = 3.0$, and $\alpha_r = 1.5$. The evaluation prompt of GPT-4o is shown in Appendix Sec. E.1.

### 6.3.2 Qualitative Results

We present qualitative comparison of image editing with other models in Fig. 6. Our approach demonstrates superior performance across diverse editing scenarios requiring semantic-spatial reasoning. Examples highlight our framework's distinctive capabilities: First, our model accurately identifies and localizes objects referenced through indirect descriptions. Second, our approach handles complex spatial instructions effectively, such as removing specific signage or adding delicate elements to precise locations. Third, our framework excels at multi-step editing operations, shown in the bottom example. The red bounding boxes visible in our results indicate the coordinates predicted by the MLLM within the GoT framework, providing interpretable insight into how our system reasons about spatial relationships during editing.

### 6.4 Ablation Study on Framework Design

We conduct an ablation study to analyze the impact of different components in our framework. Tab. 3 presents the results of our study, where we progressively integrate different components into the baseline and evaluate their effects on *GenEval* and *ImagenHub* benchmarks.

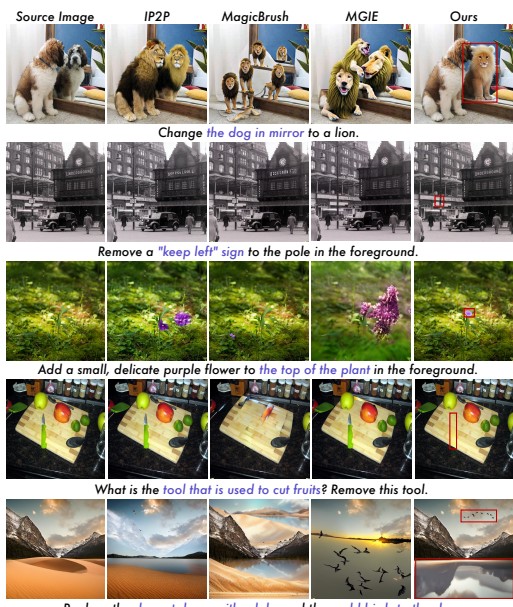

Figure 6: Qualitative results of image editing. Our GoT framework demonstrates superior performance in settings that require semantic-spatial reasoning. Red boxes indicate the coordinates predicted by MLLM within the GoT framework.

The baseline model leverages Qwen2.5-VL-3B and SDXL but does not incorporate GoT reasoning chains. It is trained with FLUX-GoT and OmniEdit-GoT for 10,000 steps. Adding GoT reasoning chains to the baseline model enables the LLM to achieve stronger semantic guidance capabilities. The reasoning process helps LLM plan for guidance in generation.

Introducing the Semantic-Spatial Guidance Module (SSGM) further enhances model performance, particularly in image editing. SSGM provides spatial control over the diffusion model, ensuring object placement aligns more accurately with the reasoning process. This

| Method | GoT | SSGM | Pretrain | GenEval | ImagenHub |
|---|---|---|---|---|---|
| Baseline | ✗ | ✗ | ✗ | 0.38 | 0.176 |
| + GoT | ✓ | ✗ | ✗ | 0.40 | 0.181 |
| + SSGM | ✓ | ✓ | ✗ | 0.42 | 0.370 |
| **GoT Framework** | ✓ | ✓ | ✓ | **0.64** | **0.533** |

Table 3: Ablation study of our GoT framework on GenEval overall and ImagenHub GPT-4o eval.

enables fine-grained editing, as reflected by the significant improvement in the ImagenHub evaluation. However, in GenEval, only the position category is notably affected by SSGM, which explains the minor performance gain.

Our final framework, which includes GoT reasoning, SSGM, and an extensive 60,000-step pretraining phase, achieves the highest scores, demonstrating the significant benefits of prolonged pretraining and the full model design. The ablation study confirms that each added component contributes positively to the overall performance, validating our framework design choices.

## 7 Conclusion

We introduced Generation Chain-of-Thought (GoT), a paradigm integrating MLLM reasoning into visual generation via explicit semantic-spatial chains, overcoming limitations in understanding object relationships. Our approach transforms visual generation from direct mapping into a reasoning-guided process with precise spatial control, addressing limitations in existing methods that lack explicit understanding of object relationships and arrangements. Leveraging large-scale datasets and a novel Semantic-Spatial Guidance Module, GoT achieves state-of-the-art, interactive visual synthesis aligned with human cognition. This advancement promises enhanced creative workflows and more controllable AI; however, the increased power to generate and manipulate visual content also demands robust ethical frameworks to mitigate misuse, such as creating misleading media.

# 8 Acknowledgement

This study was supported in part by National Key R&D Program of China Project 2022ZD0161100, in part by the Centre for Perceptual and Interactive Intelligence, a CUHK-led InnoCentre under the InnoHK initiative of the Innovation and Technology Commission of the Hong Kong Special Administrative Region Government, in part by NSFC-RGC Project N_CUHK498/24, and in part by Guangdong Basic and Applied Basic Research Foundation (No. 2023B1515130008, XW).

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

# A  Training Details

Our training process implements a two-phase approach: pretraining using LAHR-GoT, JourneyDB-GoT, and OmniEdit-GoT datasets (60,000 steps), followed by finetuning with FLUX-GoT, OmniEdit-GoT, and SEED-Edit-MultiTurn-GoT (10,000 steps). We employ low-rank adaptation (LoRA) [16] to efficiently update the Qwen2.5-VL decoder's parameters while fully optimizing the SDXL-based diffusion module. The process operates end-to-end, jointly optimizing the MLLM GoT cross-entropy token loss and diffusion MSE loss with equal weighting 1.0, demonstrating robustness without complex hyperparameter tuning. We adopt a cosine learning rate scheduler with 500 warmup steps and a maximum learning rate of $1 \times 10^{-4}$.

During the fine-tuning stage, we train the model on FLUX-GoT, OmniEdit-GoT, and SEED-Edit-MultiTurn-GoT for 10,000 steps. In this phase, we set the warmup steps to 200 and the maximum learning rate to $5 \times 10^{-5}$.

For both stages, we use the Adam optimizer with $\beta_1 = 0.9$, $\beta_2 = 0.98$, and $\epsilon = 1 \times 10^{-6}$. We also apply a weight decay of 0.05 during training. The number of batch size is set to 128.

The LLM is fine-tuned using LoRA with $r = 32$, LoRA alpha set to 32, and a LoRA dropout rate of 0.05. For diffusion, we introduce a noise offset of 0.1.

# B  Statistics of GoT T2I Datasets

The GoT T2I datasets yield rich annotations: LAHR-GoT samples with prompts averaging 110.81 characters, GoT descriptions averaging 811.56 characters, and 3.78 bounding boxes per image. Similarly, JourneyDB-GoT annotations average 149.78 characters for prompts, 906.01 characters for GoT descriptions, and 4.09 boxes image.

# C  Visualization Results

## C.1  Qualitative Analysis of Image Editing and Interactive Generation

We provide additional examples to demonstrate the capabilities of the GoT framework. Fig. 7 illustrates the image editing performance of our model. Additionally, we present the corresponding GoT content generated alongside each sample. Further examples of interactive generation using our model are shown in Fig. 8.

## C.2  Visualization of Multi-Guidance Strategy Hyperparameter Selection

We analyze the effect of hyperparameter selection in the Multi-Guidance Strategy on the generated images, as depicted in Fig. 9. The definitions of these hyperparameters are provided in Sec. 5.3.

# D  GoT Format and Examples

This section presents examples of the GoT format in our dataset. The GoT structure varies across different tasks, including text-to-image (T2I) generation, single-turn editing, and multi-turn editing.

For text-to-image generation, Fig. 10 showcases examples from FLUX-GoT, JourneyDB-GoT, and LAHR-GoT. Our GoT format represents the structured planning process of the upstream model in generating image content. It provides a detailed breakdown of the various components within an image and their spatial relationships. To enhance spatial understanding, we append location information to key objects within the GoT representation.

Fig. 11 illustrates the GoT format for image editing within our dataset. For single-turn editing, GoT represents the reasoning plan of the upstream model for a specific editing action. It consists of a description of the source image, the object to be modified, the specific editing operation, and the resulting edited image. This structured process ensures a step-by-step transformation, beginning with the original image, identifying the target object, applying the specified modification, and generating the edited image.

For multi-turn editing, GoT follows a more complex structure, as it must encapsulate the breakdown of an instruction into a sequence of consecutive steps. In practice, we first generate a description

of the source image, then decompose the multi-turn instruction into a series of step-by-step editing commands. At each step, GoT operates as a single-turn editing process, specifying the object to be modified along with the corresponding transformation. Finally, the process concludes with a description of the fully edited image.

Furthermore, for image editing tasks, positional information is appended to each object to enhance spatial comprehension.

# E  Prompts for Evaluation and Dataset Construction

### E.1  Prompts for Evaluating Image Editing Performance

We provide the prompts used for evaluating image editing performance with GPT-4o in Fig. 12. We are using GPT-4o-2024-11-20. The final score is the average of the minimum value of the two scores for each sample.

### E.2  Prompts for Text-to-Image Data Construction

Fig. 13, 14, and 16 present the key prompts utilized in text-to-image data preparation.

### E.3  Prompts for Image Editing Data Construction

Fig. 15–20 illustrate the key-step prompts employed in image editing data preparation.

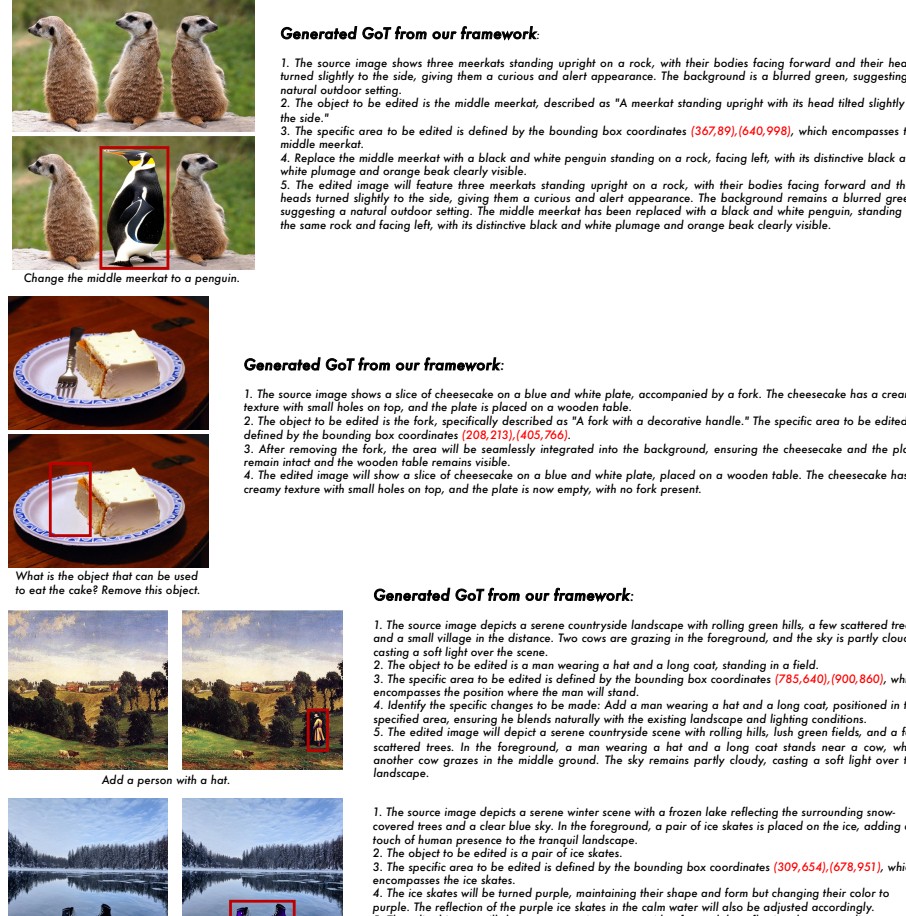

**Generated GoT from our framework:**

*1. The source image shows three meerkats standing upright on a rock, with their bodies facing forward and their heads turned slightly to the side, giving them a curious and alert appearance. The background is a blurred green, suggesting a natural outdoor setting.*
*2. The object to be edited is the middle meerkat, described as "A meerkat standing upright with its head tilted slightly to the side."*
*3. The specific area to be edited is defined by the bounding box coordinates (367,89),(640,998), which encompasses the middle meerkat.*
*4. Replace the middle meerkat with a black and white penguin standing on a rock, facing left, with its distinctive black and white plumage and orange beak clearly visible.*
*5. The edited image will feature three meerkats standing upright on a rock, with their bodies facing forward and their heads turned slightly to the side, giving them a curious and alert appearance. The background remains a blurred green, suggesting a natural outdoor setting. The middle meerkat has been replaced with a black and white penguin, standing on the same rock and facing left, with its distinctive black and white plumage and orange beak clearly visible.*

*Change the middle meerkat to a penguin.*

**Generated GoT from our framework:**

*1. The source image shows a slice of cheesecake on a blue and white plate, accompanied by a fork. The cheesecake has a creamy texture with small holes on top, and the plate is placed on a wooden table.*
*2. The object to be edited is the fork, specifically described as "A fork with a decorative handle." The specific area to be edited is defined by the bounding box coordinates (208,213),(405,766).*
*3. After removing the fork, the area will be seamlessly integrated into the background, ensuring the cheesecake and the plate remain intact and the wooden table remains visible.*
*4. The edited image will show a slice of cheesecake on a blue and white plate, placed on a wooden table. The cheesecake has a creamy texture with small holes on top, and the plate is now empty, with no fork present.*

*What is the object that can be used to eat the cake? Remove this object.*

**Generated GoT from our framework:**

*1. The source image depicts a serene countryside landscape with rolling green hills, a few scattered trees, and a small village in the distance. Two cows are grazing in the foreground, and the sky is partly cloudy, casting a soft light over the scene.*
*2. The object to be edited is a man wearing a hat and a long coat, standing in a field.*
*3. The specific area to be edited is defined by the bounding box coordinates (785,640),(900,860), which encompasses the position where the man will stand.*
*4. Identify the specific changes to be made: Add a man wearing a hat and a long coat, positioned in the specified area, ensuring he blends naturally with the existing landscape and lighting conditions.*
*5. The edited image will depict a serene countryside scene with rolling hills, lush green fields, and a few scattered trees. In the foreground, a man wearing a hat and a long coat stands near a cow, while another cow grazes in the middle ground. The sky remains partly cloudy, casting a soft light over the landscape.*

*Add a person with a hat.*

*1. The source image depicts a serene winter scene with a frozen lake reflecting the surrounding snow-covered trees and a clear blue sky. In the foreground, a pair of ice skates is placed on the ice, adding a touch of human presence to the tranquil landscape.*
*2. The object to be edited is a pair of ice skates.*
*3. The specific area to be edited is defined by the bounding box coordinates (309,654),(678,951), which encompasses the ice skates.*
*4. The ice skates will be turned purple, maintaining their shape and form but changing their color to purple. The reflection of the purple ice skates in the calm water will also be adjusted accordingly.*
*5. The edited image will show a serene winter scene with a frozen lake reflecting the surrounding snow-covered trees and a clear blue sky. In the foreground, a pair of purple ice skates is placed on the ice, adding a pop of color to the otherwise monochromatic landscape*

*Turn the color of a set of ice skates to purple*

Figure 7: More samples on image editing with the GoT content generated by our model.

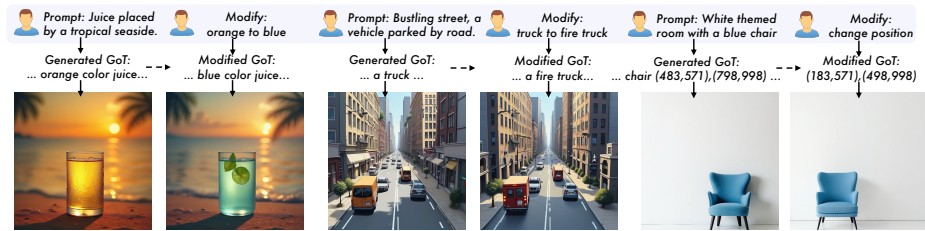

Figure 8: More examples on interactive generation.

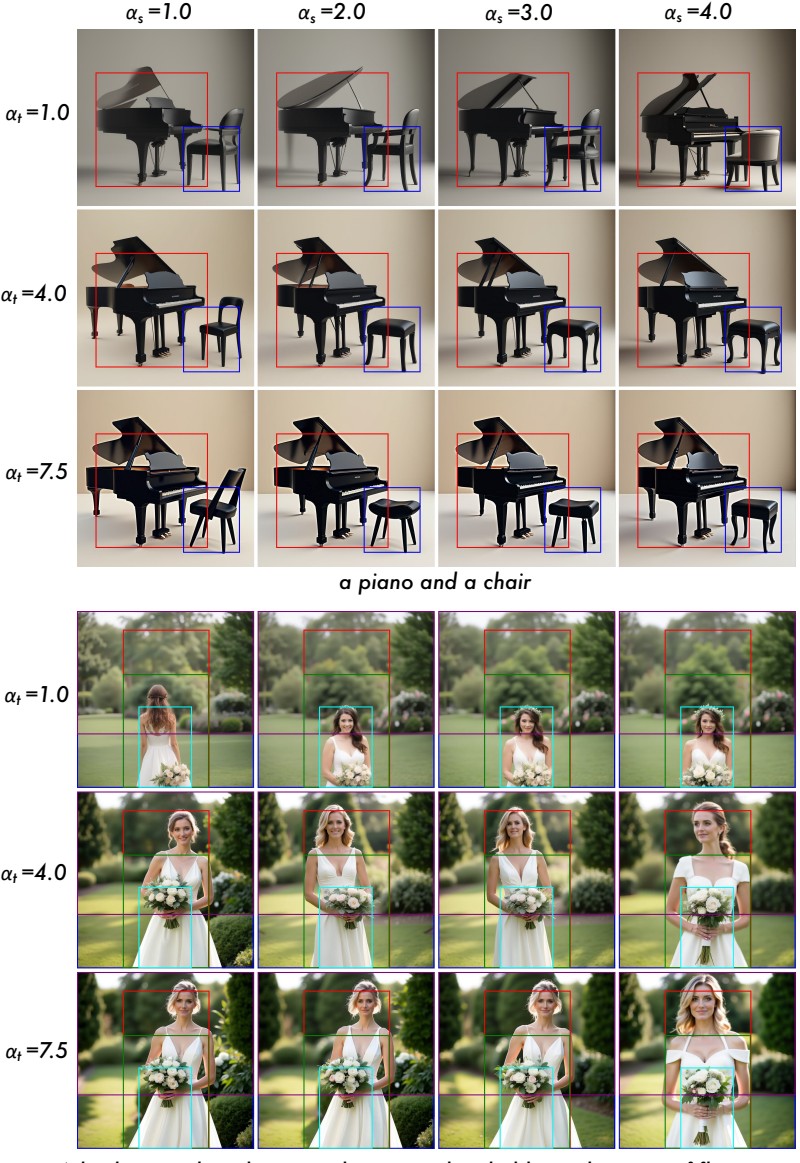

Figure 9: Visualization on Multi-Guidance Strategy Hyper-parameter Selection. The above are text-to-image samples generated by GoT framework under different hyper-parameters.

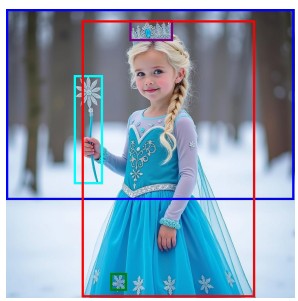

### FLUX-GoT

**Prompt:**
A young girl dressed as Elsa from Frozen, wearing a blue dress adorned with snowflake patterns, a matching crown, and holding a wand, stands in a snowy forest setting.

**GoT:**
A young girl (262,40),(862,998) stands in a snowy forest (0,0),(998,661), dressed as Elsa from Disney's "Frozen." She wears a vibrant turquoise dress adorned with intricate silver snowflake patterns (361,914),(418,975) and a matching silver belt. The dress features long sleeves and a flowing tulle skirt that cascades behind her. Her blonde hair is styled in two braids, secured with a sparkling silver crown (427,39),(578,111) featuring a large blue gem at its center. In her right hand, she holds a wand (235,229),(335,606) with a blue handle and a glittering silver star at the top, reminiscent of Elsa's magical scepter. The background is softly blurred, showcasing a serene winter landscape with snow-covered trees and a gentle light filtering through the branches, creating a magical and enchanting atmosphere.

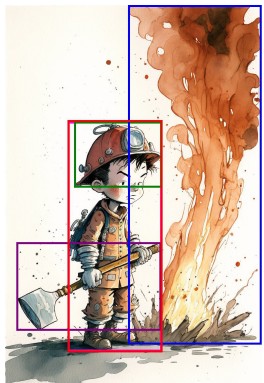

### JourneyDB-GoT

**Prompt:**
A young fireman in a helmet and uniform stands with an axe, facing a large, orange flame against a mostly white background, rendered in a hand-drawn, colorful style reminiscent of Calvin and Hobbes.

**GoT:**
The image depicts a young fireman (244,299),(613,902) standing confidently in front of a large, billowing fire (479,0),(999,882) . The fireman is dressed in a detailed, hand-drawn fireman's uniform, complete with a helmet (269,303),(611,474) , boots, and an axe (44,616),(608,846) . The uniform is rendered in a warm, earthy color palette, with the helmet featuring a prominent visor. The fire itself is a dramatic, orange-red spectacle, with smoke rising in a chaotic, blob-like form in the background. The scene is set against a mostly white background, which helps to highlight the vibrant colors and intricate details of the fire man and the fire. The overall composition is dynamic, with the fireman positioned in the foreground, ready to face the challenge of the fire, while the fire's chaotic nature adds a sense of urgency and intensity to the scene. The image is reminiscent of the style of Calvin and Hobbes, with its detailed, expressive line work and a sense of narrative tension. The use of watercolor blob in the background adds a painterly touch, enhancing the visual impact of the scene.

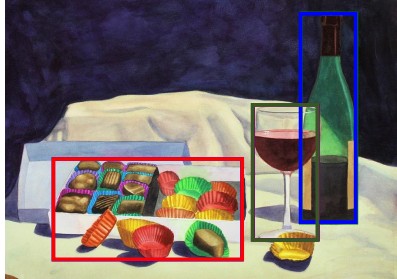

### LAHR-GoT

**Prompt:**
Still life painting featuring a bottle of wine, a glass filled with red wine, and a box of colorful chocolates on a table.

**GoT:**
The painting "Indulgence" by Cory Clifford captures a still life arrangement of luxurious confections and a glass of red wine, evoking a sense of opulence and relaxation. A box of assorted chocolates, each wrapped in vibrant foil, is open, revealing a variety of shapes and colors, including gold, green, red, and yellow. The chocolates (119,548),(608,908) are scattered around the box, some still in their wrappers, while others have been enjoyed. A bottle of red wine (751,44),(901,779) stands tall, its green glass reflecting the ambient light, accompanied by a filled wine glass (629,361),(803,838) that rests nearby. The background is a deep, rich blue, contrasting with the white tablecloth that spans the lower portion of the painting, creating a sense of depth and focus on the central objects. The overall composition is balanced, with the chocolates and wine glass positioned centrally, drawing the viewer's eye to the indulgent scene.

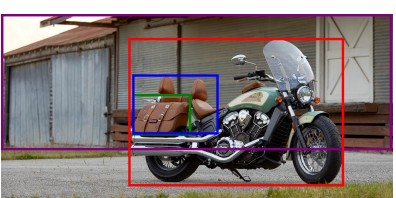

**Prompt:**
Indian Scout motorcycle with brown leather saddlebags and a green and black color scheme, parked in front of a rustic barn.

**GoT:**
The image showcases an Indian Scout motorcycle (325,123),(875,879) , prominently positioned in the foreground. The motorcycle features a sleek, vintage design with a green and cream color scheme, accented by chrome detailing. The front wheel is slightly turned to the left, and the bike is equipped with a clear windshield and a pair of round headlights. The seat is brown leather (335,309),(555,614) , complementing the saddlebags (335,409),(485,619) attached to the rear. The motorcycle is parked on a gravel surface, with a rustic barn-like structure (0,0),(997,694) in the background, characterized by corrugated metal siding and a wooden door. The overall composition places the motorcycle as the central focus, with the barn serving as a contrasting backdrop that emphasizes the bike's classic aesthetic.

Figure 10: Examples of GoT dataset for text-to-image generation, including FLUX-GoT, JourneyDB-GoT, and Laion-Aesthetics-High-Resolution-GoT.

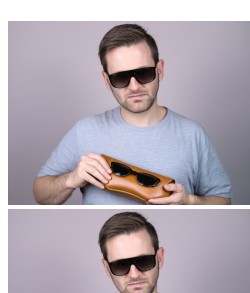

**OmniEdit-GoT**

**Instruction:**
*Turn the color of sunglasses case to purple.*

**GoT:**
1. The source image shows a person wearing a gray t-shirt and sunglasses, holding a brown leather sunglasses case. The individual appears to be in a neutral or slightly serious mood.
2. The object to be edited is the brown leather sunglasses case.
3. The specific area to be edited is defined by the bounding box coordinates (361,699),(685,996).
4. The specific changes to be made include changing the color of the sunglasses case to purple, ensuring it matches the target image description.
5. The edited image will show a person wearing a gray t-shirt and sunglasses, holding a purple sunglasses case with a pair of sunglasses inside. The background will remain unchanged but will be a solid light purple color.

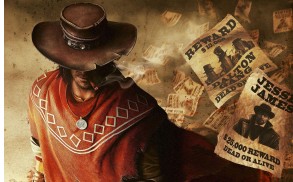
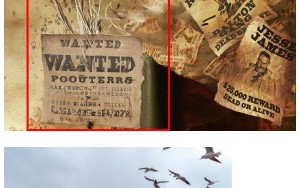

**Instruction:**
*Remove a cowboy with a brown hat, red poncho, and cigar in the foreground.*

**GoT:**
1. The source image features a cowboy with a brown hat, red poncho, and a cigar in the foreground, surrounded by old, weathered wanted posters for the Dalton Gang and Jesse James. The background is a gritty, textured wall with a vintage Western theme.
2. The object to be edited is the cowboy, specifically the one described as having a brown hat, red poncho, and cigar.
3. The specific area to be edited is defined by the bounding box coordinates (78,18),(568,999) which encompasses the entire cowboy figure.
4. Remove the cowboy completely from the image while maintaining the surrounding wanted posters and the gritty, textured background. Ensure the removal is seamless, blending the area with the old, weathered posters and the vintage Western theme.
5. The edited image will show the collection of vintage wanted posters, including those for the Dalton Gang and Jesse James, with a prominent "WANTED" sign in the foreground. The background will remain a grungy, aged wall with a sense of motion, suggesting a chaotic or historical setting. The area where the cowboy was will appear seamless, blending with the surrounding posters and background.

**SEED-Edit-MultiTurn-GoT**

**Instruction:**
*Change the red jacket on the middle to light green, remove the birds on the right, add a forest background, and put a car on the left.*

**GoT:**
1. The source image shows a person wearing a red jacket standing on a tiled walkway, surrounded by a large flock of seagulls. The scene is set against a backdrop of a body of water and a hillside with buildings, under a cloudy sky.
2. For the first step, the object to be edited is the red jacket. The specific area to be edited is defined by the bounding box coordinates (410,535),(517,687). Change the color of the red jacket to light green, and ensure the number 15 and other text remain visible.
3. For the second step, the object to be edited is the flock of birds. The specific area to be edited is defined by the bounding box coordinates (549,0),(999,553). Remove the flock of seagulls from this area.
4. For the third step, the specific area to be edited is defined by the bounding box coordinates (0,0),(999,714). Replace the background with a forest featuring a cloudy sky, a body of water, and a distant hillside with buildings. Ensure the forest creates a lush, green canopy overhead.
5. For the fourth step, the object to be edited is the left side of the image. The specific area to be edited is defined by the bounding box coordinates (0,584),(392,999). Add a car, green with a shiny, reflective surface and a sleek design, featuring a visible rear light, to this area.
6. The edited image depicts a person wearing a light green jacket with the number 15 on the back, walking down a paved path flanked by tall trees with lush green foliage. A car is parked on the left side of the path. The scene is serene and natural, with the forest creating a canopy overhead, and a cloudy sky, a body of water, and a distant hillside with buildings in the background.

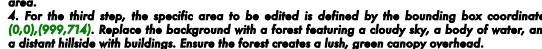
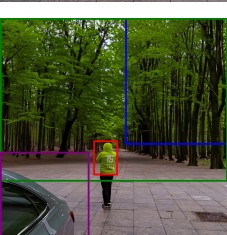

Figure 11: Examples of GoT dataset for image editing, including OmniEdit-GoT for single-turn editing and SEED-Edit-Multiturn-GoT for multi-turn editing.

**Human:**
You are a professional digital artist. You will have to evaluate the effectiveness of the AI-generated image(s) based on the given rules. You will have to give your output in this way (Keep your reasoning concise and short.): "score" : [...], "reasoning" : "..." and don't output anything else.
Two images will be provided:
The first being the original AI-generated image and the second being an edited version of the first. The objective is to evaluate how successfully the editing instruction has been executed in the second image. Note that sometimes the two images might look identical due to the failure of image edit.
From a scale 0 to 10:
A score from 0 to 10 will be given based on the success of the editing.
- 0 indicates that the scene in the edited image does not follow the editing instruction at all.
- 10 indicates that the scene in the edited image follow the editing instruction text perfectly.
- If the object in the instruction is not present in the original image at all, the score will be 0.
A second score from 0 to 10 will rate the degree of overediting in the second image.
- 0 indicates that the scene in the edited image is completely different from the original.
- 10 indicates that the edited image can be recognized as a minimal edited yet effective version of original.
Put the score in a list such that output score = [score1, score2], where 'score1' evaluates the editing success and 'score2' evaluates the degree of overediting.
Editing instruction: `<instruction>`
`<Image>` Source Image `</Image>`
`<Image>` Edited Image `</Image>`
**Assistant:**

Figure 12: Prompt for GPT4-o image editing evaluation. We are using GPT-4o-2024-11-20. The final score is the average of the minimum value of the two scores for each sample.

**Human:**
`<Image>` Image `</Image>`
You are an advanced AI visual assistant specializing in highly detailed and comprehensive visual analysis for one image. Your role is to generate a single, descriptive paragraph that encapsulates all relevant details about an image. Here is the provided image prompt for this image: `<prompt>`.
If the provided prompt aligns with the image, enhance it by adding detailed observations about the objects, their colors, shapes, textures, numeracy, and spatial relationships. If the provided prompt does not match the image content, disregard it and craft a complete description based solely on the visual elements you observe. Consider the 2D-spatial relationships (e.g., "to the left of," "near," "aligned with") and 3D-spatial relationships (e.g., "in front of," "above," "at a distance from") when describing the scene. Include details about the overall composition, highlighting how elements are arranged relative to each other, their groupings, and any complex interactions or dynamic elements within the scene. Pay close attention to the interplay of colors, textures, and shapes, ensuring that the description reflects both the visual richness and structural composition of the image. Ensure to provide the description as one single paragraph, without preamble or additional explanation.
**Assistant:**

Figure 13: Prompt for detailed recaption for text-to-image data.

**Human:**
You are tasked with identifying and extracting all the real object names from a detailed caption. An object name refers to any tangible or physical entity mentioned in the caption that can be visually grounded in the image. Ensure not to include any adjectives or single-word descriptions that do not refer to a specific object, such as "background."
Please follow these instructions:
Identify all object names in the caption in the order they appear. Maintain the exact wording of each object name as it is in the caption, including case consistency. Output the object names in a Python list format. For example, consider the following caption:
**Example 1:**
"In the image, a person is prominently featured at a vibrant pride parade, exuding confidence and pride. They are adorned in an extravagant outfit that mirrors the rainbow flag, with a deep V-neck top in bold, colorful stripes of red, orange, yellow, green, blue, and purple. The person's hair is styled in a striking rainbow color, complementing their outfit. They are surrounded by a lively crowd, with individuals wearing various colors and accessories, adding to the festive atmosphere. The background reveals a bustling street scene with buildings and trees, suggesting an urban setting. The overall composition is dynamic, with the person at the center, drawing attention to their vibrant attire and the energetic parade around them."
Your output should be a list of object names like this:
`['person', 'pride parade', 'outfit', 'V-neck top', 'The person's hair',`
`'a lively crowd', 'individuals', 'street', 'buildings', 'trees']`
**Example 2:**
"The image depicts a young boy with slender features and a pale complexion, exuding an air of arrogance and coldness. His white-blonde hair is slicked back, adding to his composed demeanor. The boy's eyes are a striking shade of cold grey, reflecting a sense of detachment and intelligence. He is dressed in a white shirt with a blue and white patterned collar, which contrasts with his pale skin and adds a touch of elegance to his appearance. The overall composition is balanced, with the boy centrally positioned against a dark background that accentuates his features and the sharpness of his expression. The interplay of colors, textures, and shapes creates a visually striking and emotionally charged image."
Your output should be a list of object names like this:
`['young boy', 'white-blonde hair', "The boy's eyes", 'white shirt']`
Now, given the following caption, extract the object names in the same format: `<caption>`
**Assistant:**

Figure 14: Prompt for identifying objects in text-to-image caption.

**Human:**
Please tell me according to the instruction: `<instruction>`. Which object is being replaced with another object? Please only answer the exact name of the two objects using the same words from the instruction. Use the format of a Python list including the two object names. The first is the 'object' and the second is the 'another_object'.
**Assistant:**

Figure 15: An example of prompt for parsing the edited object. This is used when the task type is 'replace'.

**Human:**
`<Image> Image </Image>`
Please provide the bounding box coordinates of this sentence describes: `<object_name>`
**Assistant:**

Figure 16: Prompt for grounding object. This works for both text-to-image and image editing data.

Human:
<Image> Image </Image>
You are an AI visual assistant, and you are seeing a single image. Please describe this image in one paragraph using no more than two sentences. Always remember to include describing <object_name> in the image.
Assistant:

Figure 17: Prompt for image description for image editing data.

Human:
<Image> Cropped Image </Image>
Please describe <object_name> briefly in several words no more than one sentence.
Assistant:

Figure 18: Prompt for cropped image object description for image editing.

Human:
You are a helpful visual assistant. I have an image editing data with the original instruction: <instructions>. I want to augment the instruction to obtain more free-language format instructions.
Your task is to rewrite this original instruction in English into three distinct, human-like, free-form instructions that convey the same meaning but use varied language and phrasing. The new instructions should reflect how humans might naturally request image edits.
Please provide me with three more instructions that have the same meaning as the original instruction but in a more free-language format. The new instruction can be in any format that a human might input as an editing instruction. The first instruction should be relatively concise. Use the format of a Python list which includes three items as strings.
Assistant:

Figure 19: Prompt for reinstruction for image editing data.

**Human:**
You are a helpful assistant for a designer. I have image editing data with the following information: instruction: `<instructions>`, description of source image: `<source_desc>`, description of target image: `<target_desc>`, `<coord>` `<object_desc>` Assume you are a visual assistant with access to the edit instruction and the source image. Your task is to provide a step-by-step chain of thought for the image editing process which only includes the image editing processes. The chain of thought can includes the following several type steps (can not in this order, not includes these words in the answer): Describe the source image; the object to be edited; the specific area to be edited; Identify the specific changes to be made; Describe the image after the edit. All information besides the instruction should be considered as derived from the source image. The output is meant to train a multi-modal large language model that takes the source image and instruction as input, generates the editing chain of thought, and then outputs the edited image. Therefore, your response should consider this application and provide clear, concise reasoning in numbered steps (1. 2. 3. ... etc). The response should be purely reasoning text and formatted succinctly. Ensure your answer be brief and few steps.

**In context learning, example 1:**
1. The source image shows a grand, classical building with intricate stone carvings and statues. One prominent statue, a female figure, stands on a pedestal, holding a torch and a book. The building features arched windows and a sign that reads "Learning Center."
2. The object to be edited is a statue of a woman holding a torch and a book.
3. The specific area to be edited is defined by the bounding box coordinates ((554, 166), (768, 711)), which encompasses the statue.
4. Remove the statue completely from the image while maintaining the surrounding architectural details and other elements like the building's facade, arched windows, and the "Learning Center" sign. 5. The edited image will show the grand, classical building with intricate stone carvings and the "Learning Center" sign. The statue, a female figure holding a torch and a book, will no longer be present, and the area where the statue was will appear seamless with the surrounding architecture. The building's arched windows and stone facade will remain intact.

**In context learning, example 2:**
1. The source image depicts a snowy mountain slope with a ski board in the foreground, indicating a skiing or snowboarding activity. The background features a clear blue sky and rocky terrain, suggesting a high-altitude or alpine setting.
2. The inserted object is a skier in a black jacket, complete with goggles, sitting on a snowboard. This skier will be positioned in the center of the slope, facing downhill, sitting on a snowboard.
3. The specific area to be edited is within the bounding box ((382, 303), (782, 813)), where the current object (a ski board) is located. This area needs to be replaced with the new skier.
4. The image now shows a skier dressed in a black jacket and goggles, sitting on a snowboard on a snowy slope. The background features a clear blue sky and rocky terrain, with other skiers and equipment visible in the distance. The skier is positioned in the middle of the slope, looking downhill, seamlessly blending with the existing scene.

**In context learning, example 3:**
1. The source image depicts a group of women and a child standing on a beach, all dressed in vibrant, summery outfits. The scene is bright and cheerful, with the ocean and sky forming a picturesque backdrop. The style of the image is casual and candid, capturing a moment of joy and togetherness.
2. The edited area is ((0, 0), (999, 999)), which is the whole image. The object to be edited is the group of women and the child, along with the beach and the background elements. These need to be transformed into a traditional Chinese ink painting style.
3. After the edit, the image will depict a group of women and a child standing in a traditional Chinese ink painting style, dressed in elegant, flowing garments. They will be positioned against a backdrop of serene mountains and a tranquil sea, with the overall composition reflecting the classical and detailed style of traditional Chinese ink paintings.

**Assistant:**

Figure 20: In-context assembling GoT prompt for image editing data.

