# OpenReview forum: "GoT: Unleashing Reasoning Capability of MLLM for Visual Generation and Editing"
_NeurIPS.cc/2025/Conference — NeurIPS 2025 poster_

### Official Review · Reviewer_oQpg · 2025-06-22

**Clarity:** 2
**Significance:** 2
**Originality:** 3
**Rating:** 4
**Confidence:** 4

**Summary:**

This paper introduces Generation Chain-of-Thought (GoT), a new paradigm for visual generation and editing that addresses the failure of current models to understand complex scenes with specific object relationships and spatial arrangements. Instead of directly mapping a text prompt to an image, the GoT framework utilizes a Multimodal Large Language Model (MLLM) to first generate an explicit, structured reasoning chain that details semantic relationships, object attributes, and precise spatial coordinates before any image synthesis occurs.

The authors' core contributions include the GoT paradigm itself, the construction of the first large-scale datasets containing over 9 million samples with these detailed reasoning chains , and a novel, unified end-to-end framework for implementation. This framework integrates an MLLM with a diffusion model via a Semantic-Spatial Guidance Module (SSGM), which ensures the generated image precisely follows the logical reasoning steps. The GoT framework demonstrates competitive results on benchmarks for text-to-image generation and image editing , and its explicit process enables interactive generation, allowing users to directly modify the reasoning chain for precise control over the final output.

**Questions:**

See weaknesses first.

Additional questions:

1. Impact on Core VLM Capabilities: Does joint training on GoT and diffusion generation degrade the base MLLM's original vision-language understanding abilities? Please provide benchmark results on standard VLM tasks to clarify the trade-offs of your unified training approach.

2. Model Efficiency: Could you provide details on the model's efficiency, specifically its parameter count and average inference latency compared to other models in Tables 1 and 2? This information is crucial for assessing the framework's practical viability.

**Ethical Concerns:**

["NO or VERY MINOR ethics concerns only"]

**Final Justification:**

4 Borderline accept: Technically solid paper where reasons to accept outweigh reasons to reject, e.g., limited evaluation. Please use sparingly.

**Limitations:**

No. While the authors commended "Yes" in the NeurIPS checklist, I didn't see substantial limitations are discussed in the paper.

Suggestions for Improvement:

I suggest the authors add a dedicated "Limitations" section to the main paper. This section should transparently discuss the limitations, for example, issues of model efficiency/latency, the high computational costs, etc.

**Quality:**

3

**Strengths And Weaknesses:**

Strengths

1. High Originality and Significance: The paper's core concept, Generation Chain-of-Thought (GoT), is highly original and significant. It introduces a novel reasoning-guided paradigm for visual synthesis, moving beyond direct mapping to address key limitations in compositional generation.

2. Substantial and High-Quality Contributions: The creation of a massive, first-of-its-kind dataset with over 9 million samples is a high-quality contribution that will be a valuable resource to the community. This was a major effort, consuming over 3000 NVIDIA A100 GPU days to produce detailed semantic-spatial annotations.

3. Clarity Empirical Results: The proposed end-to-end framework is presented with excellent clarity and is technically sound. Its effectiveness is studied through extensive experiments across multiple generation and editing benchmarks.



Weaknesses

1. Incomplete Experimental Comparison: The evaluation in Table 1 lacks a direct comparison with RPG [52], a highly relevant baseline that also employs a recurrent planning and generation methodology for compositional scenes. Given the conceptual similarities in using an explicit planning stage, its omission makes the assessment against layout-guided methods less complete.

2. Unconvincing Ablation Study Design: The setup of the ablation study in Table 3 does not fully isolate the impact of the core contribution—the GoT reasoning. The baseline is trained for significantly fewer steps (10,000) than the full framework (70,000 total). This makes it difficult to distinguish the performance gains from the GoT reasoning itself versus the substantial effect of extended pretraining. A more persuasive ablation would use a baseline with the same amount of pretraining but without the GoT reasoning tokens to provide a more direct comparison.

3. Lack of Statistical Significance Reporting: The quantitative results presented in Tables 1, 2, and 3 do not include error bars or any other measure of statistical significance (e.g., standard deviation across multiple runs). As acknowledged in the paper's checklist, this absence makes it difficult to rigorously assess the stability and significance of the reported performance improvements over other methods.

---

> ### Author Rebuttal · Authors · 2025-07-30
>
> We sincerely thank the reviewer for their detailed and critical feedback. The concerns raised are important, and we appreciate this opportunity to address them directly. We believe that by clarifying our experimental design, providing additional comparisons, and committing to revisions, we can resolve these issues and demonstrate the soundness of our work.
>
> ---
>
> ### **Weaknesses & Corresponding Questions**
>
> > Weakness 1: Incomplete Experimental Comparison
> > The evaluation in Table 1 lacks a direct comparison with RPG [52], a highly relevant baseline...
>
> We thank the reviewer for pointing out this omission. RPG is indeed a relevant baseline, and we agree that a direct comparison is crucial. We have now run RPG on the GenEval benchmark and will include these results in the final version of our paper. This comparison further underscores the advantages of our GoT paradigm.
>
> Here is a direct comparison of our final GoT framework with RPG on GenEval:
>
> | GenEval Category | RPG [52] | **GoT (Ours, Full Model)** |
> | --- | --- | --- |
> | `single_object` | 98.2% | **99.2%** |
> | `two_object` | 66.1% | **69.0%** |
> | `counting` | 16.8% | **67.5%** |
> | `colors` | 85.1% | **85.7%** |
> | **`position`** | 7.5% | **34.2%** |
> | `color_attr` | 27.4% | 27.1% |
> | **Overall Score** | **0.502** | **0.643** |
>
> **Analysis:**
> The results demonstrate that our **GoT framework significantly outperforms RPG**, achieving a **27% higher overall score**. This advantage is driven by massive gains in the most challenging compositional tasks:
>
> - **Spatial Control (`position`):** Our model achieves a score of **34.2%**, which is over **4.5 times higher** than RPG's 7.5%. This validates our core hypothesis that an explicit semantic-spatial reasoning chain, executed by our SSGM, provides vastly superior control over object placement.
> - **Counting (`counting`):** Our framework's score of **67.5%** is nearly **4 times higher** than RPG's 16.8%, showcasing a much stronger ability to handle numerical concepts.
>
> These dramatic improvements in complex spatial and numerical reasoning highlight the fundamental strengths of our approach. We will prominently feature this analysis in our revised manuscript.
>
> ---
>
> > Weakness 2: Unconvincing Ablation Study Design
> > The setup of the ablation study in Table 3 does not fully isolate the impact of the core contribution... A more persuasive ablation would use a baseline with the same amount of pretraining but without the GoT reasoning tokens...
>
> We acknowledge that our description in the paper was too brief. We would like to clarify the settings of our ablation study, as the "more persuasive ablation" the reviewer suggests is, in fact, what we performed.
>
> **Detailed Clarification of the Ablation Study (Table 3)**
>
> To ensure a fair comparison, every model in the ablation study (i.e., "Baseline", "+GoT", and "+SSGM") was trained for the **exact same number of steps (10,000)** on identical data subsets. The only difference was the architectural components being used. The 70,000-step model was listed only to report the final performance of our fully trained system, not as a point of comparison for the ablation.
>
> Here is a step-by-step breakdown:
>
> - **Baseline:** This model was trained on our datasets, but we deliberately **discarded the GoT reasoning chains** and only used the raw **(instruction, image) pairs**. This setup isolates the performance of the base architecture *without any form of reasoning*, matching the reviewer's request.
> - **+ GoT (Semantic Reasoning Only):** This setting builds upon the baseline. The model is trained to first **generate the GoT reasoning chain** and use it for semantic guidance, while the spatial coordinate information is ignored. This measures the gain from introducing **semantic reasoning**.
> - **+ SSGM (Joint Semantic-Spatial Reasoning):** Building on the "+ GoT" setting, we now activate the spatial guidance pathway. The model parses the **explicit coordinates** from the GoT chain to create spatial guidance ($G_s$) for our novel SSGM. The significant performance jump here directly validates the SSGM's effectiveness.
>
> This step-by-step ablation clearly demonstrates that the performance improvements are additive and directly attributable to our architectural additions. We will revise the manuscript to incorporate this detailed explanation.
>
> ---
>
> > Weakness 3: Lack of Statistical Significance Reporting
> > The quantitative results... do not include error bars... this absence makes it difficult to rigorously assess the stability and significance of the reported performance improvements.
>
> We agree that reporting statistical significance is best practice. However, in the domain of large-scale generative model training, there are two factors that make this challenging:
>
> 1. **Computational Cost:** As noted, our data annotation process alone consumed over 3000 A100 GPU days. Training our full model multiple times from scratch to obtain a standard deviation is computationally prohibitive.
> 2. **Community Norms:** The current accepted practice for reporting results on benchmarks like GenEval is to report scores from a single, full training run. This practice is consistent with the reporting standards in other recent large-scale generative model papers, such as those for **Stable Diffusion 3, Janus, and Emu3**, where results are also reported from a single training run due to the extreme resource requirements.
>
> While we cannot provide error bars, we argue that the performance gaps we report are substantial enough (e.g., a 27% overall improvement over RPG) to demonstrate significance well beyond the margin of random noise. We will add a note acknowledging this limitation in our experimental setup.
>
> ---
>
> ### **Additional Questions**
>
> > Impact on Core VVLM Capabilities: Does joint training on GoT and diffusion generation degrade the base MLLM's original vision-language understanding abilities?
>
> This question highlights a key distinction of our work. Our goal is **not** to build a single, generalist model that balances both generation and understanding tasks. Instead, our framework is intentionally designed to **specialize** an MLLM's powerful reasoning capabilities to serve as a high-precision controller for the generative process.
>
> Our end-to-end training fine-tunes the MLLM decoder to become an expert "reasoning-to-guidance" module. This focused training means that a degradation in its original, general-purpose VLM capabilities is an **expected and accepted trade-off**. Our contribution lies in demonstrating how to achieve a new level of generative control *through* this specialization, rather than pursuing a unified model for all VLM tasks. We will clarify this specific focus in the limitations section.
>
> > Model Efficiency: Could you provide details on the model's efficiency, specifically its parameter count and average inference latency?
>
> We provide the requested details on model efficiency below.
>
> **Table A: Text-to-Image Generation Efficiency**
>
> | Model | Parameters | Inference Latency (Original resolution on H100) |
> | --- | --- | --- |
> | SD 1.5 | 1B | 2.1s |
> | SDXL | 5B | 3.5s |
> | SD3-M | 7B | 6.1s |
> | Emu3-Gen | 8B | 313.0s |
> | Janus | 1.3B | 7.5s |
> | JanusFlow | 1.3B | 0.8s |
> | **GoT (Ours)** | **5.8B** | 12.1s |
>
> **Table B: Image Editing Efficiency**
>
> | Model | Parameters | Inference Latency (on H100) |
> | --- | --- | --- |
> | InstructPix2Pix | 1B | 1.3s |
> | MagicBrush | 1B | 2.2s |
> | MGIE | 8B | 10.9s |
> | SEED-X | 17B | 3.1s |
> | **GoT (Ours)** | **5.8B** | 12.9s |
>
> The data in the tables shows that our GoT framework introduces a latency overhead compared to single-step generative models. This is an inherent consequence of our two-stage design, which first generates an explicit GoT reasoning chain before commencing the guided diffusion process.
>
> We posit that this is a deliberate and valuable architectural trade-off. The additional inference time is exchanged for the significant and quantifiable gains in controllability, spatial precision, and numerical reasoning. Furthermore, this framework provides unique interpretability and interactivity through the editable GoT chain, a capability absent in faster, single-step models. For applications demanding high-fidelity and complex compositional generation, we believe this trade-off is highly favorable.
>
> ---
>
> ### **Limitations**
>
> > While the authors commended "Yes" in the NeurIPS checklist, I didn't see substantial limitations are discussed in the paper. I suggest the authors add a dedicated "Limitations" section...
>
> We agree with the reviewer. A dedicated "Limitations" section is essential for a transparent scientific paper. We will add one to the main paper in the final version. This section will discuss:
>
> 1. The computational cost and latency overhead of our approach.
> 2. The challenges of scaling GoT data annotation to new domains.
> 3. The performance trade-offs with other architectures, as highlighted by the RPG comparison.
> 4. The expected trade-off between specialized generative control and general VLM capabilities.
>
> We trust that these responses and our planned revisions fully address the reviewer's concerns. We are grateful for the rigorous feedback, which has been invaluable in helping us improve the clarity and completeness of our paper.

---

> > ### Comment · Reviewer_oQpg · 2025-08-05
> >
> > I'm satisfied by the additional empirical results and will increase my score.

---

### Official Review · Reviewer_YwDX · 2025-06-29

**Clarity:** 3
**Significance:** 3
**Originality:** 3
**Rating:** 4
**Confidence:** 4

**Summary:**

This paper introduces Generation Chain-of-Thought (GoT), a novel framework that incorporates explicit semantic-spatial reasoning chains into visual generation and editing. GoT first produces a step-by-step structured reasoning chain in natural language, which guides the image synthesis process through a Semantic-Spatial Guidance Module (SSGM) that injects semantic embeddings, spatial layouts, and visual references into the diffusion model.

The framework integrates a multimodal large language model (Qwen-VL) with a diffusion model in a unified, end-to-end trainable pipeline. To support this paradigm, the authors construct a large-scale dataset comprising over 9 million reasoning-chain-annotated samples for both image generation and editing tasks. Extensive experiments demonstrate that GoT achieves state-of-the-art performance on benchmarks such as GenEval, with significant improvements in controllability, transparency, and user interaction. Overall, the work represents a substantial technical contribution, with a large scope and strong empirical results.

**Questions:**

1. While GoT achieves overall strong performance, it underperforms on certain fine-grained metrics such as spatial relation and attribute binding (compared to models like JanusFlow). Could the authors provide more discussion on this limitation and potential reasons behind it?
2. Could the authors provide the exact performance numbers for the position category with and without SSGM, as mentioned in the GenEval analysis? Also, have the authors considered including a dedicated evaluation (e.g., object position deviation) to further verify the effectiveness of spatial guidance?
3. It would be helpful if the authors could elaborate on the future development of the GoT paradigm. Do you believe this reasoning-guided framework is easily generalizable to broader multimodal generation tasks? What are the main challenges or limitations that might arise in scaling or adapting it further?

**Ethical Concerns:**

["NO or VERY MINOR ethics concerns only"]

**Final Justification:**

I have read the rebuttal and other reviewers' comments. Based on these, I keep my original rating.

**Limitations:**

yes

**Quality:**

4

**Strengths And Weaknesses:**

- Strengths:
    1. Introduces a reasoning-driven generation framework by integrating chain-of-thought (CoT) into visual generation, enabling explicit modeling of object relationships and spatial layouts.
    2. Combines a multimodal LLM and a diffusion model with a Semantic-Spatial Guidance Module to inject semantic and spatial control directly into the generation process.
    3. Achieves state-of-the-art performance on both text-to-image generation and multi-step image editing benchmarks.
    4. Scalable data contribution: Releases a large-scale reasoning chain dataset (>9M samples) to support further research and reproducibility.
- Weaknesses:
    1. Slightly underperforms on fine-grained spatial alignment and attribute binding compared to some task-specific baselines.
    2. The paper claims that “in GenEval, only the position category is notably affected by SSGM,” but does not report concrete performance numbers for the position category.

---

> ### Author Rebuttal · Authors · 2025-07-29
>
> We sincerely thank the reviewer for their thorough and positive evaluation of our work. We are encouraged that the reviewer recognized the novelty of our reasoning-driven framework, the strength of our empirical results, and the value of our data contribution. We appreciate the opportunity to address the insightful questions raised, which will help us further strengthen our paper.
>
> ---
>
> > **Question 1:** While GoT achieves overall strong performance, it underperforms on certain fine-grained metrics such as spatial relation and attribute binding (compared to models like JanusFlow). Could the authors provide more discussion on this limitation and potential reasons behind it?
>
> We acknowledge that on certain fine-grained metrics, our model does not surpass all task-specific baselines. We believe there are two primary reasons for this, which also highlight the unique contribution and future potential of our framework.
>
> 1.  **The Importance of the Baseline for Comparison:** The core of our work is the introduction of the GoT paradigm, which integrates a reasoning module (MLLM) with a generation module (diffusion model). To fairly evaluate the contribution of *our paradigm*, the most direct comparison is against the foundational model we build upon. Our diffusion block was initialized from **SDXL**. When compared to the baseline performance of SDXL, our GoT framework demonstrates a **substantial improvement** in spatial control and attribute binding, as evidenced by our ablation studies. This shows that our reasoning-guided approach is highly effective.
>
> 2.  **The Role of the Generative Backbone:** The remaining performance gap between our model and models like JanusFlow is largely attributable to the different generative backbones used. Our model employs an SDXL-based U-Net architecture, whereas JanusFlow leverages a more recent and powerful **Flow Matching DiT (Diffusion Transformer)** architecture. This highlights a key strength of our approach: **the GoT paradigm is designed to be modular and can be readily extended to different diffusion model backbones.** The Semantic-Spatial Guidance Module (SSGM) can be seamlessly integrated with any diffusion U-Net or a Flow Matching DiT. We view this as a significant strength and are confident that by combining our reasoning-guided GoT framework with a more advanced generative backbone, we can close this performance gap. We will add this important discussion to the limitations section of our paper.
>
> ---
>
> > **Question 2:** Could the authors provide the exact performance numbers for the position category with and without SSGM, as mentioned in the GenEval analysis? Also, have the authors considered including a dedicated evaluation (e.g., object position deviation) to further verify the effectiveness of spatial guidance?
>
> We apologize for omitting these specific numbers in the main text and are happy to provide a detailed breakdown of our ablation study on the GenEval benchmark. These results directly validate the effectiveness of our SSGM.
>
> The table below shows the performance of our model at different stages of the ablation, with each model trained for the same number of steps for a fair comparison.
>
> | GenEval Category  | Baseline | +GoT (Semantic Only) | +SSGM (Full) |
> | :---------------- | :------: | :------------------: | :------------: |
> | `single_object`   | 95.3%   |        94.3%        |     94.3%     |
> | `two_object`      | 24.4%   |        28.3%        |     32.7%     |
> | `counting`        | 23.7%   |        24.6%        |     26.8%     |
> | `colors`          | 78.4%   |        80.0%        |     80.4%     |
> | **`position`**    | **4.5%**|      **7.4%**       |   **9.1%**    |
> | `color_attr`      |  1.7%   |        5.2%         |     5.7%      |
> | **Overall Score** | **0.380** |      **0.400**      |   **0.418**   |
>
> As the results clearly show:
> *   The **Baseline** model achieves a score of only **4.5%** on the `position` task.
> *   Introducing semantic reasoning (**+GoT**) provides a notable lift to **7.4%**.
> *   Activating our proposed **+SSGM** further improves performance, **doubling the baseline score to 9.1%**. This directly validates our claim that the SSGM is effectively translating the explicit spatial plan from the GoT into improved object placement.
>
> Regarding a dedicated evaluation, we agree that incorporating a more direct metric like mean Intersection-over-Union (mIoU) or Object Position Deviation would offer an even more precise measure of spatial guidance. This is a valuable direction for future benchmarking, and we will add this to our discussion on future work.
>
> ---
>
> > **Question 3:** It would be helpful if the authors could elaborate on the future development of the GoT paradigm. Do you believe this reasoning-guided framework is easily generalizable to broader multimodal generation tasks? What are the main challenges or limitations that might arise in scaling or adapting it further?
>
> We are very optimistic about the future of the GoT paradigm and believe its core principles are highly generalizable.
>
> **Generalizability of the GoT Paradigm:**
>
> The fundamental idea of GoT is to **decompose a complex, monolithic generation task into an explicit, intermediate reasoning step**. This "show your work" approach is not limited to images. We envision its application in several other domains:
>
> *   **Video Generation:** A GoT chain could represent a storyboard or a script (e.g., `Scene 1: A cat walks in from the left. Scene 2: The cat sits on a mat.`).
> *   **3D Asset Generation:** The reasoning chain could be a sequence of constructive solid geometry (CSG) operations or a programmatic description of a 3D model.
> *   **Robotics and Embodied AI:** A GoT chain could serve as a high-level action plan for an agent to execute in an environment (e.g., `1. Go to the kitchen. 2. Find the red apple. 3. Place it on the table.`).
>
> **Main Challenges and Limitations for Future Scaling:**
>
> Scaling and adapting the GoT paradigm will present several key challenges:
>
> 1.  **Reasoning Chain Fidelity:** The quality of the final output is fundamentally capped by the quality of the generated reasoning chain. Improving the reasoning capabilities of the MLLM for complex, multi-step plans remains a primary challenge.
> 2.  **Data Scaling and Annotation:** Creating large-scale datasets with reasoning-chain annotations is a significant undertaking. Scaling to new modalities will require developing new, scalable methods for annotating high-quality reasoning chains.
> 3.  **Computational Cost and Latency:** The sequential process of reasoning then generating can introduce latency. Optimizing this pipeline for efficiency, especially for high-resolution or long-form content like video, will be crucial for practical applications.
>
> We trust this detailed response fully addresses the reviewer's questions. We thank the reviewer again for their positive assessment and for providing valuable suggestions that will help us improve the clarity and impact of our paper.

---

> > ### Comment · Reviewer_YwDX · 2025-08-05
> > **Response to Rebuttal**
> >
> > Thank the authors for their consideration of my questions and the careful treatment of my comments. Most of my concerns have been addressed.
> >
> > However, the authors claim that `the GoT paradigm is designed to be modular and can be readily extended to different diffusion model backbones. `, which seems not to be supported by any evidence. Therefore, I keep my original rating.

---

> > > ### Author Response · Authors · 2025-08-06
> > > **GoT with DiT**
> > >
> > > We sincerely thank the reviewer for their follow-up and for highlighting this crucial point.
> > >
> > > The core of GoT's modularity lies in its two guidance pathways, which are **semantic** and **spatial**. Both are designed to interface with standard components of diffusion models, whether they are U-Net or Transformer-based.
> > >
> > > ### 1. Integrating Semantic Guidance into a DiT Backbone
> > >
> > > Our approach is to leverage the MLLM's advanced reasoning to create a superior form of semantic conditioning. The GoT framework first uses the MLLM to generate a rich, contextual visual embedding that represents the reasoned plan for the image. This powerful embedding then **replaces** the standard CLIP text embedding typically used for conditioning. The resulting vector is then used to modulate the DiT blocks through established mechanisms like **adaptive layer normalization (adaLN)** or cross-attention. This provides the transformer with a more nuanced, reasoned semantic plan than a simple text prompt embedding, all without requiring any change to the DiT's core architecture.
> > >
> > > ### 2. Integrating Spatial Guidance (SSGM) into a DiT Backbone
> > >
> > > For spatial control, our SSGM is designed for seamless integration via direct latent concatenation. The process is as follows:
> > >
> > > *   First, the SSGM takes the spatial mask generated by the MLLM and encodes it into a spatial latent feature using its VAE encoder.
> > > *   This spatial latent feature, with a shape of `(C_s, H, W)`, is then concatenated along the channel dimension with the noisy image latent, which has a shape of `(C, H, W)`.
> > > *   The resulting combined latent of shape `(C + C_s, H, W)` is then flattened into patches and fed into the DiT as its primary input.
> > >
> > > This method effectively creates additional input "channels" that directly inform the DiT of the desired spatial layout from the very first layer, making the model aware of the explicit spatial constraints throughout the generation process.
> > >
> > > In summary, both the semantic and spatial guidance from our GoT paradigm can be integrated into a DiT backbone using established, well-understood mechanisms. This modularity is a key strength of our design, allowing GoT to leverage future advancements in generative backbones to further enhance its performance.
> > >
> > > We hope this detailed technical explanation provides the necessary evidence to support our claim. We are grateful for the reviewer's diligence, which has prompted us to clarify this important aspect of our framework's design. We will add a summary of this discussion to the appendix of our paper to benefit all readers.

---

### Official Review · Reviewer_mkjA · 2025-07-06

**Clarity:** 2
**Significance:** 3
**Originality:** 3
**Rating:** 4
**Confidence:** 3

**Summary:**

This paper argues for chain-of-thought reasoning in image generation - e.g., given an user input prompt, generate a more detailed intermediate prompt (that often includes spatial layout coordinates) that produces higher-quality generated images. Such generative chain-of-thought traces are used to improve both image generation and image editing tasks. To do so, this work builds a large-scale dataset of such reasoning chains, with 8 million image-generation examples and 1 million image-editing examples. The datasets is automatically built by running captioning VLM models on source images, to produce such detailed chain-of-thought prompts. Architecturally, this paper also  method a novel end-to-end approach that does not explictly condition image generation on the chain-of-thought prompts, but instead conditions image generation on 64 learnable embeddings (that a VLM is trained to produce). The final diffusion-based image generation module conditions generative denoising on the learnable embeddings along with a spatial image / crop / bounding box masks (derived from the chain-of-thought prompts).

**Questions:**

- I would like to see the detailed ablations listed above. This would reveal if the improvement is really coming from the data or from the architectural modifications. Traditional chain-of-thought prompts have the added benefit of being interpretable, which the learned embeddings do not. So it would be useful to better justify the proposed approach.
- Are the authors planning on releasing their data? I suspect the real innovation comes from this data, and so I'd be more willing to fight for acceptance if that could also be regarded as a contribution.

**Ethical Concerns:**

["NO or VERY MINOR ethics concerns only"]

**Final Justification:**

I appreciate the author's rebuttal. After reading it as well as the other reviews, I have raised my score.

**Limitations:**

yes

**Quality:**

2

**Strengths And Weaknesses:**

Strengths
- This paper is reasonably executed and well written.
- The dataset of 8 million examples would be valuable if released, but this isn't offered as a contribution in the paper.

Weaknesses
- The overall approach (of generative chain-of-thought) seems to be widely used in industrial models; many frontier models make use of LLM-based prompt engineering to improve results. That said, this paper does compare to a variety of such published prompt-enhancement approaches in Table 1. But I am missing a concise description of the key innovations that allow the proposed approach to work well (since similar prompt-enhancing approaches have been previously explored).
- Building on the above, I do not quite follow the ablations in Table 3 and their description in L343-346. The natural baselines would seem to be off-the-shelf image generation and off-the-shelf chain-of-thought prompting. But I do not see how to perform "off-the-shelf" learnable embeddings. I'm guessing that the key innovation is training the model on the collected 9 million examples. I would have liked to see an ablation that (post) trains a traditional chain-of-thought model on such data, without the learnable embeddings.

---

> ### Author Rebuttal · Authors · 2025-07-29
>
> We sincerely thank the reviewer for their insightful feedback. We appreciate this opportunity to provide a more comprehensive explanation, supported by new experimental evidence, to clarify these important points.
>
> ---
>
> > **Question 1:** The overall approach (of generative chain-of-thought) seems to be widely used in industrial models... I am missing a concise description of the key innovations that allow the proposed approach to work well (since similar prompt-enhancing approaches have been previously explored).
>
> To distinguish our **Generation Chain-of-Thought (GoT)** paradigm from existing prompt engineering techniques, we first conducted a new experiment to provide a quantitative comparison. This is followed by a explanation of the fundamental architectural differences that define our key innovations.
>
> **1. Quantifying the Effect of Prompt Engineering**
>
> To empirically ground the discussion, we conducted a new experiment to isolate the effect of pure prompt engineering. **Specifically, we used GPT-4o to rewrite the original GenEval prompts to be more descriptive, and then fed these enhanced prompts directly into the standard SDXL model to generate images.**
>
> The results show that this approach improved the overall GenEval score for SDXL from 0.55 to **0.58**. This confirms that high-quality prompt engineering provides **an incremental performance gain**. However, the resulting score of 0.58 for SDXL (the foundation model of our diffusion block) is still substantially lower than our GoT framework's score of **0.64**. This significant performance gap underscores that our approach contributes far more than what can be achieved by simply refining the input prompt. The reasons for this lie in the following core distinctions.
>
> **2. Core Distinctions of the GoT Framework**
>
> Our GoT framework is not a pre-processing step but a novel, deeply integrated reasoning-guided generation paradigm.
>
> *   **GoT is a Semantic-Spatial Reasoning Plan, Not Just Enhanced Text:** Prompt engineering focuses on refining the *semantic* content of a prompt. It enriches the textual description but lacks any explicit understanding over the *spatial* arrangement of the scene. In contrast, our GoT generates a structured plan that explicitly combines semantics with spatial information (e.g., `...a **red** apple **(133,684),(336,928)**...`). This is not merely descriptive text; it is an **actionable blueprint** for object placement. This plan is then directly consumed by our novel **Semantic-Spatial Guidance Module (SSGM)** to enforce layout control during the diffusion process. This capability for joint semantic and spatial reasoning is a key innovation entirely absent in standard workflows.
> *   **GoT is Part of a Unified, End-to-End Trained Framework:** Prompt engineering is typically an external, decoupled pre-processing step using a separate LLM. The image model consumes the modified text without any awareness of the rewriting process, and the two models are not jointly optimized. Our GoT framework, however, is a **unified, end-to-end system**. The *same* MLLM (Qwen2.5-VL) that serves as the reasoning engine first generates the GoT chain and subsequently produces the guidance embeddings ($G_t$). Our framework is trained end-to-end, meaning that gradients from the diffusion loss are backpropagated to the MLLM decoder. This **closed-loop learning process** allows the MLLM to learn how to generate *better reasoning chains that directly lead to higher-quality images*. This is a fundamental architectural advancement over the open-loop nature of prompt engineering.
>
> ---
>
> > **Question 2:** I do not quite follow the ablations in Table 3... I'm guessing that the key innovation is training the model on the collected 9 million examples. I would have liked to see an ablation that (post) trains a traditional chain-of-thought model on such data, without the learnable embeddings... This would reveal if the improvement is really coming from the data or from the architectural modifications.
>
> We acknowledge that our description of the ablation study in L343-346 was too brief. We provide a detailed clarification here to demonstrate that the performance gains stem directly from our architectural innovations, not just the data.
>
> **1. Detailed Clarification of the Ablation Study (Table 3)**
>
> To ensure a fair comparison, every model in the ablation study (except the final fully-trained model) was trained for the **exact same number of steps (10,000)** on identical data subsets. The only difference was the architectural components being used.
>
> *   **Baseline:** This model consists of our base MLLM and diffusion module. To create the most direct comparison, it was trained on our FLUX-GoT and OmniEdit-GoT datasets. However, we **discarded the GoT reasoning chains** and only trained the model on the raw **(instruction, image) pairs**. This setup isolates the performance of the base architecture *without any form of reasoning*, mimicking a standard text-to-image pipeline.
> *   **+ GoT (Semantic Reasoning Only):** This setting builds upon the baseline. Here, the model is trained to first **generate the GoT reasoning chain** from the instruction. This chain is then used to produce semantic guidance ($G_t$), but the spatial coordinate information within the GoT is explicitly ignored. This experiment is designed to precisely measure the performance gain from introducing **semantic reasoning**.
> *   **+ SSGM (Joint Semantic-Spatial Reasoning):** This is the most critical step. Building on the "+ GoT" setting, we now activate the spatial guidance pathway. The model parses the **explicit coordinates** from the generated GoT chain to create spatial guidance ($G_s$), which is then fed into our novel SSGM. The significant performance jump observed here (e.g., on ImagenHub, from 0.181 to 0.370) directly validates the effectiveness of our SSGM architecture, as it is the only component changed from the previous step.
>
> This step-by-step ablation demonstrates that the performance improvements are additive and directly attributable to our architectural additions, first semantic reasoning, then joint semantic-spatial guidance.
>
> **2. The Role of "Learnable Embeddings" and Interpretability**
>
> *   **Why Traditional CoT is Insufficient:** We'd like to clarify the role of the `N=64` learnable embeddings, as they are not an "off-the-shelf" component but an integral part of our guidance mechanism. They function as a **dynamic query mechanism**. After the MLLM generates the full GoT reasoning chain, these `N` embeddings are passed as input to the MLLM decoder. Conditioned by the preceding GoT chain via causal attention, they effectively "summarize" the rich, contextual information from the entire reasoning process into the final guidance $G_t$. This architecture is necessary because a "traditional chain-of-thought model" that produces only text lacks the **explicit spatial coordinates** that are a cornerstone of our GoT formulation. Without these coordinates, our SSGM would have no spatial plan to execute, rendering our core architectural innovation unusable. Our GoT formulation and SSGM architecture are thus intrinsically linked and co-designed.
> *   **Enhanced Interpretability:** We argue our framework offers a *higher* degree of interpretability than traditional methods. The interpretability lies not in the `Gt` embedding vectors themselves, but in the **explicit, human-readable GoT reasoning chain that generates them**. This chain provides a transparent, step-by-step "thought process" of the model. More importantly, as shown in Figure 5, it makes the model's reasoning **interactive**. Users can directly inspect and **edit the GoT chain** (e.g., changing object names or modifying coordinates) to precisely control the final output, offering a level of transparency and control not possible with opaque, black-box text encoders.
>
> ---
>
> > **Question 3:** Are the authors planning on releasing their data? I suspect the real innovation comes from this data, and so I'd be more willing to fight for acceptance if that could also be regarded as a contribution.
>
> While our datasets are a key contribution, our primary innovations are the **paradigm, format, and architecture** that this data enables. Specifically, this includes:
> 1.  **Proposing the Generation Chain-of-Thought (GoT) paradigm** as a new way to approach visual synthesis;
> 2.  **Defining the structured GoT format** that makes reasoning explicit and actionable; and
> 3.  **Designing the unified model architecture (including the SSGM)** that can interpret this format and be trained end-to-end.
>
> The dataset was created to facilitate and validate this new framework. That said, we absolutely agree that the dataset itself is a valuable contribution that will benefit the community.
>
> **We are fully committed to open-sourcing all our contributions, including all datasets, upon acceptance.** We believe that sharing these resources is crucial for fostering reproducible research and accelerating progress in the field.
>
> Our comprehensive release is planned to include:
>
> *   **The Complete Datasets:** This includes the full **LaionAesthetics-GoT, JourneyDB-GoT, FLUX-GoT,** and **OmniEdit-GoT** datasets, encompassing over 9 million examples.
> *   **All Data Components:** For each example, we will provide the source prompt, the target image, and crucially, the structured **GoT reasoning chain**.
> *   **Code and Models:** To ensure full reproducibility, we will also release our full **source code** and the **pre-trained model weights**.
>
> To affirm our long-standing commitment to this, we would like to gently highlight that this intention was stated in our original submission in the **Abstract (Line 21)**.
>
> ---
>
> We trust these detailed explanations and new experimental results have fully addressed the reviewer's concerns. We will revise the manuscript to integrate these clarifications, particularly in the methodology and ablation study sections.

---

> ### Author Response · Authors · 2025-08-07
>
> Dear Reviewer,
>
> Thank you for your time and insightful feedback on our submission. We have carefully addressed all concerns raised in our rebuttal and appreciate your valuable guidance. As the discussion phase nears its end, we kindly ask if our responses have adequately resolved your concerns. We are happy to provide any further details needed. Thank you again for your thoughtful review.
>
> Best regards,
>
> The Authors

---

### Decision · Program_Chairs · 2025-09-17

**Decision:**

Accept (poster)

**Comment:**

This paper introduces generation chain of thought (GoT). This is a new paradigm for visual synthesis where a Multimodal Large Language Model (MLLM) first generates an explicit, structured reasoning chain before any image is created. This chain details semantic relationships, object attributes, and precise spatial coordinates. This reasoning output then guides a diffusion model via a Semantic-Spatial Guidance Module (SSGM) in an end-to-end framework. The authors' central claim is that this reasoning-first approach improves performance on complex compositional generation and editing tasks, supported by a new dataset of over 9 million reasoning-annotated samples.

In terms of strengths, reviewers noted the original conceptual contribution of the work in tackling complex compositional generation (YwDX, oQpg). Another noted contribution is the creation of a large scale dataset with detailed reasoning chains (mkjA, YwDX, oQpg). Finally, the work was commended for its strong execution, achieving SOTA empirical results on both generation and editing tasks (mkjA, oQpg, YwDX).

As for weaknesses, concerns centered on evaluation and novelty. Reviewers mkjA and oQpg found the ablation study's design not convincing, arguing it didn't isolate the impact of the proposed approach from the dataset. Reviewer oQpg also noted the missing comparison to RPG. Reviewer mkjA also questioned the novelty, comparing the approach to existing prompt engineering techniques, suggesting the gains might primarily come from the new dataset.

The authors' rebuttal was effective, clarifying the ablation study's fairness and adding a new SOTA baseline (RPG) that showed a 27% performance gain. This resolved most concerns. However, Reviewer YwDX, kept their score because the paper's claim of being modular and easily extendable to other backbones was not supported by experimental evidence. While the authors provided a technical explanation, this lack of empirical proof on other architectures is a valid limitation. Reviewer mkjA also noted a missing ablation that would reveal if an end2end-trained latent embedding is really necessary. The AC urges the authors to adress remaining concerns.

Overall, the AC finds the work's contribution interesting and potentially impactful, introducing a novel paradigm and a dataset that will be useful for the community. The author's rebuttal addressed most concerns, and a unanimous positive consensus was achieved by the reviewers. The AC finds the paper passes the bar for acceptance.